



**Modeling long-term fire impact on ecosystem characteristics and surface energy using a**

**process-based vegetation-fire model SSiB4/TRIFFID-Fire v1.0**

**Huilin Huang[1], Yongkang Xue[1, 2], Fang Li[3], and Ye Liu[1]**

1. Department of Geography, University of California, Los Angeles, CA 90095, USA

2. Department of Atmospheric & Oceanic Sciences, University of California, Los Angeles, CA

90095, USA

3. International Center for Climate and Environmental Sciences, Institute of Atmospheric Physics,

Chinese Academy of Sciences, Beijing, China

Correspondence to: Yongkang Xue (yxue@geog.ucla.edu)





## Abstract

Fire is one of the primary disturbances to the distribution and ecological properties of the world's major biomes and can influence the surface fluxes and climate through vegetation-climate interactions. This study incorporates a fire model of intermediate complexity to a biophysical model with dynamic vegetation, SSiB4/TRIFFID (The Simplified Simple Biosphere Model coupled with the Top-down Representation of Interactive Foliage and Flora Including Dynamics

Model). This new model, SSiB4/TRIFFID-Fire, updating fire impact on the terrestrial carbon cycle every 10 days, is then used to simulate the burned area during 1948-2014. The simulated global burned area in 2000-2014 is 471.9 Mha yr$^{-1}$, close to the estimate, 478.1 Mha yr$^{-1}$, in Global Fire Emission Database v4s (GFED4s) with a spatial correlation of 0.8. The SSiB4/TRIFFID-Fire reproduces temporal variations of the burned area at monthly to interannual scales. Specifically, it

captures the observed decline trend in northern African savanna fire and accurately simulates the fire seasonality in most major fire regions. The simulated fire carbon emission is 2.19 Pg yr$^{-1}$, slightly higher than the GFED4s (2.07 Pg yr$^{-1}$).

      The SSiB4/TRIFFID-Fire is applied to assess long-term fire impact on ecosystem characteristics and surface energy budget by comparing model runs with and without fire (FIRE-

ON minus FIRE-OFF). The FIRE-ON simulation reduces tree cover over 6.14% of the global land surface, accompanied by a decrease in leaf area index and vegetation height by 0.13 m$^2$ m$^{-2}$ and 1.27 m, respectively. The surface albedo and sensible heat are reduced throughout the year, while latent heat flux decreases in the fire season but increases in the rainy season. Fire results in an increase in surface temperature over most fire regions.





## 1. Introduction

Wildfire, whether natural or human-made, is one of the primary ecosystem disturbances and it plays a major role in the terrestrial biogeochemical cycles and ecological succession across spatial and temporal scales (Sousa, 1984; Bond-Lamberty et al., 2007; Bowman et al., 2009). Every year in the dry season, wildfires burn about 400 Mha of land vegetated areas, leaving behind

numerous scars in the landscape (Randerson et al., 2012; Giglio et al., 2013; Chuvieco et al., 2016). Fires affect the climate through modification of water, energy, and momentum exchange between land and atmosphere (Chambers and Chapin, 2002; Bond-Lamberty et al., 2009; Li and Lawrence, 2017; Li et al., 2017) and can interact with monsoons by affecting atmospheric circulations (De Sales et al., 2016; Saha et al., 2016; De Sales et al., 2018). Fires are also important sources of

global carbon, aerosols, and trace gas emissions. Based on the latest satellite estimates, global fires emit $1.5 - 4.2$ Pg C $yr^{-1}$ carbon, $7 - 8.2$ Pg C $yr^{-1}$ $CO_2$, and $1.9 - 6.0$ Tg C $yr^{-1}$ black carbon to the atmosphere (Kaiser et al., 2012; Darmenov and da Silva, 2013; Chuvieco et al., 2016; van der Werf et al., 2017; Li et al., 2019). Fire emissions contribute to increases in greenhouse gases and cloud condensation nuclei through geochemistry processes (Scholes et al., 1996; Andreae and Merlet,

2001), affecting radiative forcing, hydrology cycle (Ward et al., 2012; Jiang et al., 2016; Hamilton et al., 2018), and air quality (van der Werf et al., 2010; Johnston et al., 2012).

Since the early 2000s, fire models have been developed within Dynamic Global Vegetation Models (DGVMs) to explicitly describe the burned area, fire emissions, and fire disturbance on terrestrial ecosystems (Thonicke et al., 2001; Venevsky et al., 2002; Arora and Boer, 2005;

Thonicke et al., 2010; Li et al., 2012; Pfeiffer et al., 2013; Lasslop et al., 2014; Yue et al., 2014; Rabin et al., 2018; Burton et al., 2019; Venevsky et al., 2019). These fire models have various levels of complexity, from simple statistical models (SIMFIRE; Knorr et al., 2016) to complicated



process-based ones such as SPITFIRE (Thonicke et al., 2010) and MC2 (Bachelet et al., 2015).

With increasing complexity, more fire processes and fire characteristics are considered in fire

models. In general, current fire models broadly capture the global amounts and spatial distribution

of burned area and carbon emissions, as compared to different observations. However, many

empirically determined parameters are included in the complicated process-based models, which

leads to large uncertainties. There is no model that outperforms other models across all fire

variables (Hantson et al., 2020). Moreover, current fire models have deficiencies in simulating the

peak fire month, fire season length, and interannual variability, as reported by the Fire Model

Intercomparison Project (FireMIP; Hantson et al., 2020; Li et al., 2019). Most fire models show a

1-2 months shift in peak burned area and simulate a longer fire season compared to observations.

Fire models have been used to reconstruct fire history before the satellite era (Kloster et al.,

2010; Yang et al., 2015; van Marle et al., 2017; Li et al., 2019). In addition, they are widely used

to attribute historical variability of burned area to various climate and anthropogenic driving

factors (Kloster et al., 2012; Andela et al., 2017; Chaste et al., 2018; Forkel et al., 2019; Teckentrup

et al., 2019). Some fire models have been used to assess long-term fire impact on the terrestrial

carbon cycle by comparing a reference simulation with fire and a sensitivity simulation

representing "a world without fire." However, the simulated responses of vegetation and carbon

cycle are divergent. Bond et al. (2005) reported that forest cover would double in a world without

fire with a less constrained burned area, while a smaller tree cover increase of 15 – 38% is

suggested by recent fire-coupled DGVMs (Lasslop et al., submitted). Earlier model-based studies

reported that fire reduced terrestrial carbon uptake. However, the range of the quantified reduction

was fairly broad (0.05–3.60 Pg C yr$^{-1}$), and most studies did not consider the fire effects on





vegetation distribution and related mechanisms (Li et al., 2014; Yue et al., 2015; Poulter et al.,

2015; Yang et al., 2015; Seo and Kim, 2019; Zou et al. 2020).

Thus far, only fire model developed by Li et al. (2012; 2013) has been used to investigate

the long-term fire effects on surface energy (Li et al., 2017). By comparing the simulated climate

with and without fire, this study concluded that fire caused a significant decrease in surface

radiation, latent heat, and sensible heat fluxes through changes in biophysical properties such as

albedo, Bowen ratio, and aerodynamic resistance. An increase in surface temperature was found

over most fire regions. In Li et al. (2017), however, only annual energy fluxes changes were

quantified, and the impact of fire on vegetation distribution was not taken into account.

In the original SSiB4/TRIFFID, the carbon disturbance caused by fire and insects was

assumed to be a constant which depended solely on plant functional type (PFT) without spatial

and temporal changes (Cox et al., 2001; Liu et al., 2019). However, the fire disturbance is highly

varied with climate, vegetation productivity, and socio-economic conditions, which has a strong

influence on vegetation dynamics, carbon cycling, and soil processes. In this study, we develop

the fire modeling by incorporating fire scheme of Li et al. (2012; 2013) to SSiB4/TRIFFID

(hereafter SSiB4/TRIFFID-Fire). The SSiB4/TRIFFID-Fire model updates fire-induced carbon

loss every 10 days, which has been rarely employed in current process-based fire models, and is

used to provide a quantitative assessment of fire impact on ecosystem characteristics and surface

energy at subseasonal, seasonal, interannual, and long-term scales. Specifically, our objectives are:

(1) to evaluate the climatology and interannual variability of burned area and carbon emissions

simulated by offline SSiB4/TRIFFID-Fire; (2) to assess the ability of SSiB4/TRIFFID-Fire in

capturing the fire seasonality in major fire regions; and (3) to assess the long-term fire impact on

PFT distribution and vegetation properties and the resultant changes in surface energy budget and



temperature. In Sect. 2, we provide a brief description of the DGVM, SSiB4/TRIFFID, and the

fire model, Li et al. (2012; 2013), and the coupling procedures. The experimental design and data

for model input and validation are introduced in Sect. 3. The fire model evaluation on the global

scale and the application of long-term fire impact on the ecosystem characteristics and surface

properties are presented in Sect. 4. Discussions and conclusions are given in Sect. 5.

## 2. Method

### 2.1 Land and vegetation model

The Simplified Simple Biosphere Model (SSiB, Xue et al., 1991; Zhan et al., 2003) is a

biophysical model which simulates fluxes of surface radiation, momentum, sensible/latent heat,

runoff, soil moisture, surface temperature, and vegetation gross/net primary products (GPP/NPP)

based on energy and water balance. The SSiB was coupled with a dynamic vegetation model, the

Top-down Representation of Interactive Foliage and Flora Including Dynamics Model (TRIFFID),

to calculate leaf area index (LAI), canopy height, and PFT fractional coverage according to the

carbon balance (Cox, 2001; Zhang et al., 2015; Harper et al., 2016; Liu et al., 2019). We have

improved the PFT competition strategy and plant physiology processes to make the

SSiB4/TRIFFID suitable for seasonal, interannual, and decadal studies (Zhang et al., 2015; Liu et

al., 2019). SSiB4/TRIFFID includes seven PFTs: (1) broadleaf evergreen trees (BET), (2)

needleleaf evergreen trees (NET), (3) broadleaf deciduous trees (BDT), (4) C3 grasses, (5) C4

plants, (6) shrubs, and (7) tundra. The coverage of a PFT is determined by net carbon availability,

competition between species, and disturbance, which implicitly includes mortality due to fires,

pests, and windthrow. A detailed description and validation of SSiB4/TRIFFID can be found in

Zhang et al. (2015) and Liu et al. (2019).





## 2.2 Fire model and modification

In this study, a process-based fire model of intermediate complexity has been implemented in the SSiB4/TRIFFID, called SSiB4/TRIFFID-Fire. The fire model developed by Li et al. (2012; 2013) was first built on the model platform of CLM-DGVM, and has been incorporated in IAP-DGVM (Zeng et al., 2014), CLM4.5 (Oleson et al., 2013), CLM5 (Lawrence et al., 2019), LM3 in Earth system model GFDL-ESM (Rabin et al., 2018; Ward et al., 2018), AVIM in Climate System Model BCC-CSM (W. P. Li, personal comm.), E3SM Land Model (ELM; Ricciuto et al., 2018), NASA GEOS catchment-CN4.5 model (Zeng et al., 2019), and DLEM (Yang et al., 2014), and partly used in GLASS-CTEM (Melton and Arora, 2016). The following briefly describes the fire schemes adapted from Li et al. (2012; 2013) and Li and Lawrence (2017) and our modifications.

The fire model comprises three parts: fire occurrence, fire spread, and fire impact. The basic equation is that the burned area in a grid cell ($A_b$, km$^2$ s$^{-1}$) is determined by the number of fires per time step ($N_f$, count s$^{-1}$) and the average spread area per fire ($a$, km$^2$ count$^{-1}$):

$$A_b = N_f a. \tag{1}$$

### 2.2.1 Fire occurrence

$N_f$ is the product of the number of potential ignition counts due to both natural causes and human activities ($I_i$, count s$^{-1}$ km$^{-2}$), fuel availability ($f_b$), fuel combustibility ($f_m$), and human suppression factor ($f_{eo}$). In this paper, we only consider non-crop fire by excluding the cropland fraction ($f_{crop}$) from burning:

$$N_f = I_i f_b f_m f_{eo} (1 - f_{crop}) A_g, \tag{2}$$

where $A_g$ is the land area of the grid cell (km$^2$). Fires in the croplands are excluded here due to their small extent within the major fire regions and their relatively low intensity. Cropland fire is still a major uncertainty in remote sensing datasets (Randerson et al., 2012) and more data and





investigation are needed.

The number of potential ignitions ($I_i$, count s$^{-1}$ km$^{-2}$) is composed of both natural and
anthropogenic ignitions, which are parameterized following Li et al. (2012), Prentice and Mackerras (1977), and Venevsky et al. (2002). The fuel availability $f_b$ (fraction, range 0−1) is given as:

$$f_b = \begin{cases} 0 & B_{ag} \leq B_{low} \\ \frac{B_{ag}-B_{low}}{B_{up}-B_{low}} & B_{low} < B_{ag} < B_{up}, \\ 1 & B_{ag} \geq B_{up} \end{cases} \tag{3}$$

where $B_{ag}$ (g C m$^{-2}$) is the aboveground biomass (leaf and stem in SSiB4/TRIFFID-Fire) of all
PFTs. Following Li et al. (2012), we use $B_{low}$ = 155 g C m$^{-2}$ as the lower fuel threshold, below which fire does not occur and $B_{up}$ = 1050 g C m$^{-2}$ as the upper fuel threshold, above which fuel load is not a constraint for fire occurrence.

Fuel combustibility $f_m$ (fraction, 0−1) is estimated following Eq. (4):

$$f_m = f_{RH}f_\theta, \tag{4}$$

where $f_{RH}$ and $f_\theta$ represent the dependence of fuel combustibility on relative humidity (RH; %) and the root zone soil moisture ($\theta$), respectively (Li and Lawrence, 2017). Following Li et al. (2013), we assume $f_m$= 0 when surface air temperature $T$ is below -10 °C. $f_{RH}$ reflects the impact of real-time climate conditions on fuel combustibility while $f_\theta$ reflects the response of fuel combustibility to preceding climate conditions (Shinoda and Yamaguchi, 2003):

$$f_{RH} = \begin{cases} 0 & RH \leq RH_{low} \\ (\frac{RH_{up}-RH}{RH_{up}-RH_{low}})^{1.3} & RH_{low} < RH < RH_{up}, \\ 1 & RH \geq RH_{low} \end{cases} \tag{5}$$

$$f_\theta = \begin{cases} 0 & \theta \leq \theta_{low} \\ (\frac{\theta_{up}-\theta}{\theta_{up}-\theta_{low}})^{0.7} & \theta_{low} < \theta < \theta_{up}. \\ 1 & \theta \geq \theta_{up} \end{cases} \tag{6}$$





Relative humidity suppresses fire occurrence when it is larger than $RH_{up}$ = 70 %, and relative humidity does not constrain fire when it is smaller than $RH_{low}$ = 30 %. The $\theta_{up}$ and $\theta_{low}$ are used as the upper and lower thresholds of soil moisture in a similar way as the thresholds of relative humidity. The parameters vary with PFTs, as shown in Table S1.

The human suppression factor ($f_{eo}$; 0-1) reflects the demographic ($f_d$) and economic ($f_e$) impact on fire occurrence in populated areas ($D_p > 0.1$ per person km⁻²):

$$f_{seo} = f_d f_e. \tag{7}$$

It is assumed that people do not play a role in fire suppression ($f_{seo}$=1) when population density is smaller than 0.1 person km⁻². A detailed description of $f_d$ and $f_e$ parameterization can be found in Li et al. (2012; 2013).

### 2.2.2 Average spread area after fire ignition

The average spread area of a fire is assumed elliptical in shape with the ignition point located at one of the foci and the fastest spread occurring along the major axis. The average burned area of a fire $a$ (km² per count) without human intervention is represented as (Li et al., 2012):

$$a = \pi L_B (u_{max} g_0 F_m \tau / 1000)^2 F_{se}, \tag{8}$$

where $L_B$ is the length-to-breadth ratio of the ellipse shape and is related to the wind speed, $W$ (m s⁻¹):

$$L_B = 1.0 + 10.0[1 - \exp(-0.06W)]. \tag{9}$$

$u_{max}$ is the PFT-dependent maximum fire spread rate (m s⁻¹;Table S2); $g_0 = 0.05$ is the dependence of fire spread rate perpendicular to the wind direction; $F_m$ is the influence of fuel wetness on fire spread and is related to relative humidity and root zone soil moisture. Following Li and Lawrence (2017), $F_m$ is assumed to be related to $f_m$ in the fire occurrence Eq. (4):

$$F_m = f_m^{0.5}. \tag{10}$$



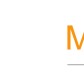 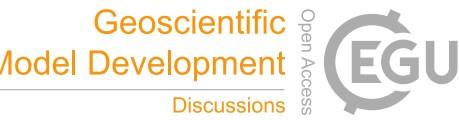

$\tau$ (=3600*24 s) is the global average fire duration, which is simply taken to be one day as reported

by Giglio et al. (2006). The human suppression factor, $F_{se}$, reflects human impact on fire spread

through fire-fighting activities and is parameterized following Li et al. (2013).

**2.2.3 Carbon emissions, post-fire mortality, and emissions of aerosols and trace gases**

     In post-fire regions, the fire carbon emission, $\varphi_j$ (g C s$^{-1}$), from the vegetation tissue (leaf,

stem, root) of the $j$th PFT is calculated based on the burned area ($A_b$; km$^2$ s$^{-1}$):

$$\varphi_j = A_b \boldsymbol{C}_j \cdot \boldsymbol{CC}_j. \tag{11}$$

$\boldsymbol{C_j} = (C_{leaf}, C_{stem}, C_{root})$ is carbon density vector (g C km$^{-2}$) for leaf, stem, and root of the $j$th

PFT calculated in TRIFFID; $\boldsymbol{CC_j}$ is the corresponding combustion completeness for leaf, stem, and

root of the $j$th PFT (Table S3). The total carbon emission from a grid cell, $\varphi$ (g C s$^{-1}$), is taken as

the weighted average of emission from each PFT by its fractional coverage ($f_j$):

$$\varphi = \sum \varphi_j f_j. \tag{12}$$

     Meanwhile, fire-induced mortality transfers carbon from uncombusted leaf, stem, and root

to litter:

$$\psi_j = A_b \boldsymbol{C}_j \cdot (1 - \boldsymbol{CC}_j) \cdot \boldsymbol{M}_j, \tag{13}$$

where $\boldsymbol{M_j} = (M_{leaf}, M_{stem}, M_{root})$ is the corresponding mortality factor (Table S3).

     Finally, the emissions of trace gases and aerosols species $x$ for the $j$th PFT ($EM_{x,j}$, g s$^{-1}$)

can be calculated from carbon emissions ($\varphi_j$) using the PFT-dependent emission factor ($EF_{x,j}$, g

species (kg dm)$^{-1}$):

$$EM_{x,j} = EF_{x,j} \frac{\varphi_j}{[C]}, \tag{14}$$

where $[C]$ (= 0.5 g C (kg dm)$^{-1}$) is a unit conversion factor from dry matter to carbon (Li et al.,

2019). The emission factors, $EF_{x,j}$, of trace gases and aerosols in Table S4 are based on Andreae





(2019). The emissions of trace gases and aerosols can be applied in the atmospheric chemistry model to calculate the production of secondary aerosols, transport of pollutants, and the resultant aerosol direct and indirect effects on climate.

**2.3 Implementing fire model in SSiB4/TRIFFID**

**2.3.1 Coupling procedures**

In SSiB4/TRIFFID, SSiB4 provides GPP, autotrophic respiration, and other physical variables such as canopy temperature and soil moisture every 3 hours for TRIFFID (Fig. 1). TRIFFID accumulates the 3-hourly GPP and respiration and provides biotic carbon, PFT fractional

coverage, vegetation height, and LAI every 10 days, which are used to update surface properties (albedo, canopy height, roughness length, and aerodynamic/canopy resistances) in SSiB4. When the fire model is included, it uses the meteorological forcings and physical variables provided by SSiB4 every 3 hours and the biophysical properties (vegetation fraction and biotic carbon) provided by TRIFFID every 10 days. The fire model calculates the burned area, carbon combustion,

post-fire mortality, and emissions every 3 hours, and the fire-induced carbon loss is subtracted every 3 hours from fuel load. The carbon loss is accumulated within 10 days in the fire model and is transferred to TRIFFID on Day 10. TRIFFID updates the vegetation dynamics based on carbon balance on Day 10, using the net primary production, fire-induced carbon loss, and PFT competition strategy. The updated vegetation dynamics are transferred to SSiB4 to reflect the fire

impact on surface properties.



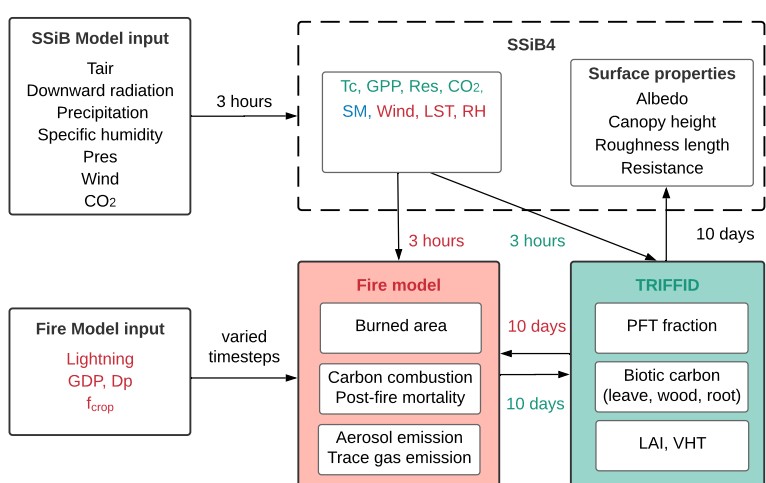

\* Red color: inputs for fire model; Green color: inputs for TRIFFID; Blue
color: inputs for both Fire and TRIFFID

**Figure 1.** Schematic diagram of fire model coupling in SSiB4/TRIFFID-Fire (Tair: air temperature;
Pres: surface pressure; Dp: population density; $f_{crop}$: crop fraction; Tc: canopy temperature; Res:
autotrophic respiration; SM: soil moisture; LST: land surface temperature; RH: relative humidity;
VHT: vegetation height)

### 2.3.2 Including the fire effect on the carbon pool

In TRIFFID (Cox, 2001), the fractional change of the $j$th PFT ($\frac{df_j}{dt}$) is governed by the

Lotka–Volterra equation:

$$\frac{df_j}{dt} = \frac{\lambda_j \, NPP_j \, f_j}{Cv_j} \left[1 - \sum_j c_{ij} f_j\right] - \gamma_j f_j, \tag{15}$$

where $f_j$ is the fractional coverage of the $j$th PFT, $\lambda_j \, NPP_j$ is the carbon available for spreading,

$Cv_j$ is the carbon density (g C km$^{-2}$), $c_{ij}$ is the competition coefficient between the $i$th and $j$th PFTs,

and $\gamma_j$ (s$^{-1}$) is the constant disturbance representing the loss of PFT fraction due to fires, pests,

windthrow, and many other processes.

In SSiB4/TRIFFID-Fire, the loss of PFT fraction due to fires ($\beta_j$) can be explicitly derived

from the fire-induced carbon loss:





$$\beta_j = \frac{(\varphi_j + \psi_j) \cdot f_j}{Cv_j}, \tag{16}$$

where $\varphi_j$ and $\psi_j$ are PFT-dependent carbon loss due to combustion and post-fire mortality in Eq. (11) and Eq. (13), respectively. The fire-caused PFT fraction loss results in bare soil for vegetation spreading decided by the competition strategy in TRIFFID. As such, fire disturbance is explicitly

represented which varies in space and time, and the original $\gamma_j$ is adjusted to $\gamma_j'$ to exclude fire disturbance (Table 1):

$$\frac{df_j}{dt} = \frac{\lambda_j \cdot NPP_j \cdot f_j}{c_j} \Big[ 1 - \sum_j c_{ij} f_j \Big] - \big( \gamma_j' + \beta_i \big) f_j. \tag{17}$$

A similar explicit approach of fire disturbance on PFT fraction is also applied in JULES-INFERNO (Burton et al., 2019).


**Table 1.** The disturbance rate implicitly including fire disturbance ($\gamma_v$) and excluding fire disturbance ($\gamma_v'$).

|  | BET | NET | BDT | C3 grasses | C4 plants | Shrubs | Tundra |
|---|---|---|---|---|---|---|---|
| $\gamma_v$ | 0.004 (0.04 with grasses) | 0.004(0.04 with grasses) | 0.004 (0.04 with grasses) | 0.1 | 0.1 | 0.05 | 0.05 |
| $\gamma_v'$ | 0.004 | 0.004 | 0.004 | 0.02 | 0.02 | 0.04 | 0.01 |

## 3. Experimental setup and data

**3.1 Experimental design**

Two sets of offline experiments have been conducted using SSiB4/TRIFFID-Fire, which consist of FIRE-ON (SSiB4/TRIFFID-Fire with fire model switched on) and FIRE-OFF (SSiB4/TRIFFID-Fire with fire model switched off). To obtain the initial conditions for these two experiments, similar to our previous SSiB4/TRIFFID experiments (Zhang et al., 2015; Liu et al.,

2019), we conducted spin-up simulations (SP$_{FIRE-ON}$ and SP$_{FIRE-OFF}$) to reach a quasi-equilibrium



PFT distribution with and without fire disturbance. These spin-up simulations were initialized using the quasi-equilibrium state from Liu et al. (2019, SP$_{INIT}$ in Fig. 2) and were driven by climatology forcing averaged in 1948-1972 and atmospheric $CO_2$ concentration, population density, and GDP in 1948 (Fig. 2).

Based on the quasi-equilibrium status with fire disturbance (SP$_{FIRE-ON}$), a transient run was performed (FIRE-ON) with the fire model turned on from 1948 to 2014 (Fig. 2). The model was forced by 3-hourly meteorological forcings, yearly updated atmospheric $CO_2$ concentration, population density, and GDP data. FIRE-ON produced the fire regime, ecosystem, and surface conditions during 1948-2014. A FIRE-OFF run, based on SP$_{FIRE-OFF}$, was carried out with the fire

model switched off during 1948-2014. The vegetation distribution was allowed to respond to climate variations in both FIRE-ON and FIRE-OFF simulations and to fire disturbances only in the FIRE-ON.

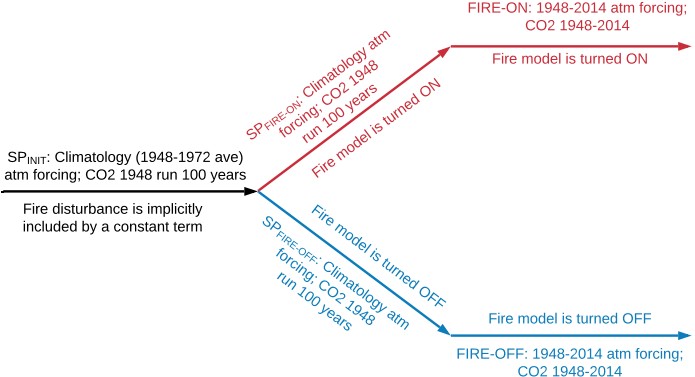

**Figure 2.** Experiment design for FIRE-ON and FIRE-OFF experiments

**3.2 Model input and validation data**

The meteorological forcings used to drive SSiB4/TRIFFID-Fire for the period of 1948–2014 are from the Princeton global meteorological dataset for land surface modeling (Sheffield et al., 2006), including surface air temperature, surface pressure, specific humidity, wind speed,





downward shortwave radiation flux, downward longwave radiation flux, and precipitation (Table

2). The dataset is constructed by combining global observation-based datasets with the

NCEP/NCAR reanalysis. The spatial resolution is 1.0°1.0° and the temporal interval is 3 hours.

The required inputs for driving the fire model are listed in Table 2. The 2-hourly

climatology lightning data from NASA LIS/OTD v2.2 at 2.5°2.5° resolution is used to calculate

the natural ignitions for fire occurrence. The population density data are provided by the Gridded

Population of the World version 3 (GPWv3; CIESIN, 2014) for 1990-2005 and Database of the

Global Environment version 3.1 (HYDEv3.1; Goldewijk et al., 2010) for 1850−1980. GDP per

capita in 2000 is from van Vuuren et al. (2007). The population density and GDP data are used to

calculate the human ignitions and suppression in the fire model. The agriculture fraction is

obtained from the GLC2000 (Bartholome and Belward, 2005). All the datasets are resampled to

1.0°1.0° spatial and 3-hourly temporal resolution.

**Table 2.** Datasets used to drive SSIB4/TRIFFID-Fire and evaluate simulations

| Variables | Sources | Resolution |
|---|---|---|
| Surface air temperature<br>Surface pressure<br>Specific humidity<br>Wind speed<br>Downward shortwave radiation<br>Downward longwave radiation<br>Precipitation | Sheffield et al., (2006) | 1°, 3-hourly |
| Lightning frequency | NASA LIS/OTD v2.2 | 2.5°, 2-hourly |
| Population density | GPWv3 (CIESIN, 2005);<br>HYDE v3.1 (Klein Goldewijk et al., 2010) | 0.5°, 5 yearly<br>5', 10 yearly |
| Gross domestic product (GDP) | van Vuuren et al. (2006) | 0.5°, in 2000 |
| Agriculture fraction | GLC2000 (Bartholome et al., 2002) | 1°, in 2000 |
| Burned area<br>Fire emission | GFED4s (Randerson et al., 2012;<br>van der Werf et al. 2017) | 0.25°, monthly |
| GPP | FLUXNET-MTE (Jung et al. 2009) | 0.5°, monthly |





The Global Fire Emission Database (GFED) is a fire dataset derived mainly from MODIS

satellite observations (van der Werf et al., 2006; van der Werf et al., 2010; Giglio et al., 2013).

The GFED fire product provides the burned area and fire emissions on the global scale and has

been widely used for fire model validation and calibration (van Marle et al., 2017; Li et al., 2019).

The latest version of GFED, GFED4s, has included the contribution from small fires that are below

the MODIS detection limit (Randerson et al., 2012; Giglio et al., 2013; van der Werf et al., 2017).

The burned area and carbon emissions simulated by SSiB4/TRIFFID-Fire will be validated using

gridded monthly GFED4s fire products in 2000–2014 at 0.25° spatial resolution.

FLUXNET Model Tree Ensemble (FLUXNET-MTE) GPP is upscaled from FLUXNET

observations to the global scale using the machine learning technique MTE (Jung et al., 2011).

The FLUXNET-MTE GPP at 0.5°0.5° spatial resolution in 1982–2011 has been resampled to

1.0°1.0° to be compared with simulated GPP in SSiB4/TRIFFID-Fire.

## 4. Results

This section evaluates the model simulation of burned area, carbon emissions, and GPP by

comparing FIRE-ON results with GFED4s and FLUXNET-MTE data. Specifically, we will focus

on the model description of fire seasonality. After model validation, SSiB4/TRIFFID-Fire is

applied to assess the long-term fire effect on the ecosystem and climate using the differences

between the FIRE-ON and FIRE-OFF.

### 4.1 Burned Area

The simulations of burned area are evaluated using satellite-based product GFED4s for the

period of 2000-2014. Figure 3 shows the 2000-2014 annual burned fraction in GFED4s and

SSiB4/TRIFFID-Fire and their latitudinal distribution. The simulated global burned area is 471.9





Mha yr$^{-1}$, slightly higher than the estimate from MODIS Collection 6 in 2002-2016 (423 Mha yr$^{-1}$; Giglio et al., 2018) but very close to the value from GFED4s (478.1 Mha yr$^{-1}$). The spatial distribution of observed burned area is well captured in the SSiB4/TRIFFID-Fire simulation with a spatial correlation of 0.80. Both GFED4s and SSiB4/TRIFFID-FIRE show that the major burned

area is concentrated in the tropical savannas (5-15º N; 5-20º S), including the Northern Hemisphere African (NHAF), Southern Hemisphere African (SHAF), Southern Hemisphere South American (SHSA), and northern Australia (Figs. 3a and 3c). GFED4s shows that the Northern African savanna has a larger latitudinal burned area in a narrower fire band compared to Southern African savanna (Fig. 3b). SSiB4/TRIFFID-Fire captures the latitudinal band in burned Northern African

savanna but underestimates its magnitude (Fig. 3d). Another burned area peak that occurs around 50º N is also simulated where the high levels of aboveground biomass in Boreal Asia (BOAS) and Boreal North America (BONA) facilitate extreme fire occurrence while the humid climate there actually suppresses fire ignition (Yue et al., 2014). The low burned fraction for deserts and tropical rainforests, which is caused respectively by low fuel availability and combustibility, are also well

simulated (Fig. 3a), leading to the minimum burned area around the equator and subtropical regions.

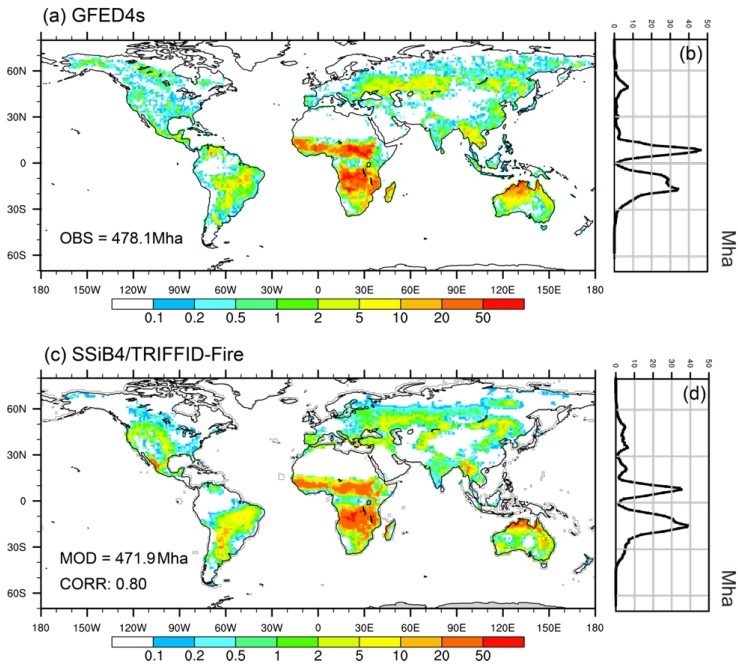

**Figure 3.** Spatial distribution of annual burned fraction (%) averaged over 2000–2014 for (a) GFED4s and (c) SSiB4/TRIFFID-Fire. The right panel shows zonal mean burned area (Mha) for (b) GFED4s and (d) SSiB4/TRIFFID-Fire

We also evaluate the 14 sub-regions following the definition in GFED according to the fire behavior similarity (van der Werf et al., 2006). The burned area in African savanna accounts for more than 60 % of the global burned area in both GFED4s and SSiB4/TRIFFID-Fire (Fig. 4b). The simulated burned areas in SHAF and NHAF are 168.3 Mha and 124.5 Mha, respectively, very close to GFED4s burned areas in SHAF (167.9 Mha) but slightly underestimated in NHAF (155.5 Mha). The negative bias in NHAF burned area is the main cause of the underestimation in the latitudinal fire distribution around 10° N (Fig. 3d). SSiB4/TRIFFID-Fire also captures the burned fraction in other major fire regions such as Australia (AUST) and SHSA (Fig. 4b), which are dominated by savanna fire. We notice that the burned area in western and central parts of temperate North America (TENA) is overestimated (Figs. 3c and 4b). Nevertheless, the burned area there is



relatively small compared to that in major fire regions such as SHAF and NHAF. This shortcoming has been reported in a number of fire models (Pfeiffer and Kaplan, 2012; Lasslop et al., 2014; Yue et al., 2014; Venevsky et al., 2019). The reasons include an underestimate of anthropogenic suppression, inaccurate description of fuel pattern/grassland fraction, and landscape fragmentation

from roads and other anthropogenic features. The burned area in the Middle East (MIDE) is also overestimated (Fig. 4b) as a larger burned area is simulated at the northern boundary of the Sahara Desert and south of the Black Sea (Fig. 3c). The simulated burned area is underestimated in BONA and BOAS (Fig. 4b) where fire has a lower incidence but a longer duration compared to the global average (Ward et al., 2018; Venevsky et al., 2019). As we assume all fires persist for one day, the

burned area in boreal regions is therefore underestimated. Further improvements, such as multi-day burning and a deliberate scheme for anthropogenic effect, are necessary in regional applications.

        In 8 out of the 14 sub-regions, SSiB4/TRIFFID-Fire well reproduces the observed interannual variability (IAV) of burned area, with the correlation between simulations and

observations significant at $p < 0.05$ (Fig. 4c). The regions are NHAF, SHSA, AUST, TENA, Central America (CEAM), Europe (EURO), Southeast Asia (SEAS), and Equatorial Asia (EQAS). In particular, a decline in NHAF burned area is found in both SSiB4/TRIFFID-Fire and GFED4s, which has been attributed to agricultural expansion and intensification in recent fire studies (Andela et al., 2017; Teckentrup et al., 2019). Although our model does not have an explicit

description of agriculture fraction and intensification changes, the anthropogenic effect is implicitly included by relating fire suppression to population density and GDP (Li et al., 2013). Meanwhile, SSiB4/TRIFFID-Fire also captures the IAV in SEAS and EQAS, which is known to be driven by climate factors such as relative humidity and soil moisture.





**Figure 4.** (a) Map of 14 regions used in this study, after Giglio et al. (2006, 2010) and van der Werf et al. (2006, 2010) (b) Annual burned area (Mha) averaged over 2000-2014 for GFED4s and SSiB4/TRIFFID-Fire in 14 regions (c) Annual burned area (Mha) for 2000–2014 for GFED4s and SSiB4/TRIFFID-Fire in 14 GFED regions. The "*" indicates the positive correlation is significant at $p < 0.05$.

The simulated IAV of SHAF burned area is not as good as other savanna fire regions (e.g.,

NHAF, SHSA, and AUST), although the IAV is small there (Fig. 4c). Some studies have reported





that humans have a substantial impact on SHAF fire, which limits the effect of climate-induced IAV (Archibald et al., 2010; Venevsky et al., 2019). In addition, the simulated IAV of burned areas

is lower than observations in BONA and BOAS (Figs. 4c) as the model fails to capture some extreme fire events (Fig. S1). The lower variability comes from the climatology lightning data (Pfeiffer et al., 2013). As lightning flash is the predominant ignition source in the Northern Hemisphere high latitudes, the application of climatology lightning has a greater impact in boreal regions than in other parts of the globe.

Figure 5 shows the pointwise temporal correlation of the multi-year monthly burned area between SSiB4/TRIFFID-Fire and GFED4s averaged in 2000-2014. SSiB4/TRIFFID-Fire captures the fire seasonality in most regions, including Southern African savanna, South American savanna, the northeastern part of Boreal Asia, the eastern part of Boreal America, Southeast Asia, and Equatorial Asia. Specifically, we examine the simulation of peak fire month and fire season

length in SSiB4/TRIFFID-Fire following the definition that fire seasons include months with more than 1/12 of the mean annual burned fraction (Venevsky et al., 2019). Over the globe, August and December are the two peak fire months that have the largest contribution to the annual burned area (Fig. 6a). SSiB4/TRIFFID-Fire generates two fire seasons in June-July-August and December-January-February, capturing the peak month in August but underestimating the burned area in

December. In the tropical savannas (SHAF, SHSA, and NHAF), fire activities concentrate in the local dry season, and the burned area during the fire season accounts for more than 80 % of the annual burned area (Figs. 6b-d). The burned area in Southern Hemisphere major fire regions, SHAF and SHSA, peaks in August and September in both observations and model. The simulated fire seasons in SHAF (June-October) and SHSA (July-October) match precisely with the

observations. In other fire regions such as SEAS and EQAS, SSiB4/TRIFFID-Fire also reproduces





the fire seasonality and peak fire months (Figs. 6e-f). Compared with the latest results from other fire models (Hantson et al., 2020), our model produces more realistic burned area peak and fire season duration.

Figure 5 shows, however, that the fire seasonality in West Africa, the western part of Boreal
Asia, eastern China, western Australia, and the middle eastern US needs to be improved. The simulated fire season in West Africa is December-March (Fig. 6d), slightly shifted from the fire season in GFED4s (November-February), which contributes to the lower fire peak in December in the global burned area. The recent FireMIP models also show a 2-months delay in peak fire month in Northern tropics (Hantson et al., 2020), which might be related to the representation of
seasonality in vegetation production and fuel build up. In western Siberia (Fig. 6g), the largest burned area is observed in April, which is the wettest month in our model that comes from the high precipitation and specific humidity in the forcing data. A similar scenario is found in Western Australia (Fig. 6h). Meanwhile, the absence of crop fire in SSiB4/TRIFFID-Fire also contributes to the low temporal correlation with the observations in agricultural areas such as the middle
eastern US and eastern China.

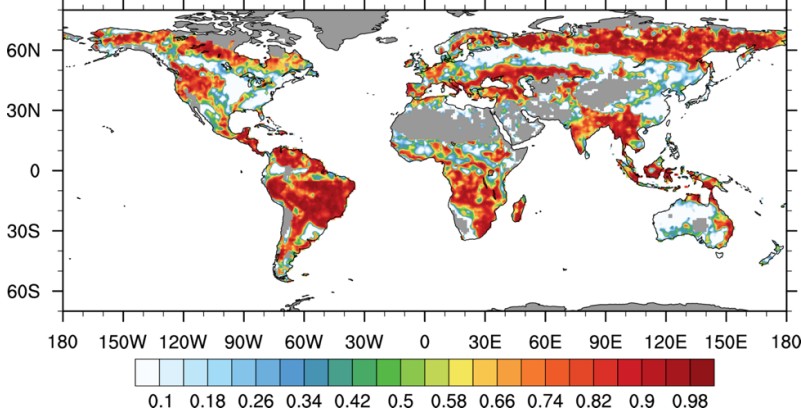

**Figure 5.** Temporal correlation of monthly burned area averaged over 2000-2014 between SSiB4/TRIFFID-Fire and GFED4s (grids with annual burned fraction < 0.01 % are masked)





Overall, SSiB4/TRIFFID shows good consistency in the simulation of peak fire month and

fire season duration in most regions, probably related to the better representation of vegetation-

fire interactions in SSiB4/TRIFFID-Fire which updates fire effects on vegetation dynamics every

10 days. The inaccurate simulation of fire season in several fire regions could come from

deficiency of the forcing data, the inaccuracy in dynamic vegetation processes, or some processes

that control the fire but are not represented in the model. More comprehensive observational data

are needed to improve the simulation in these areas.

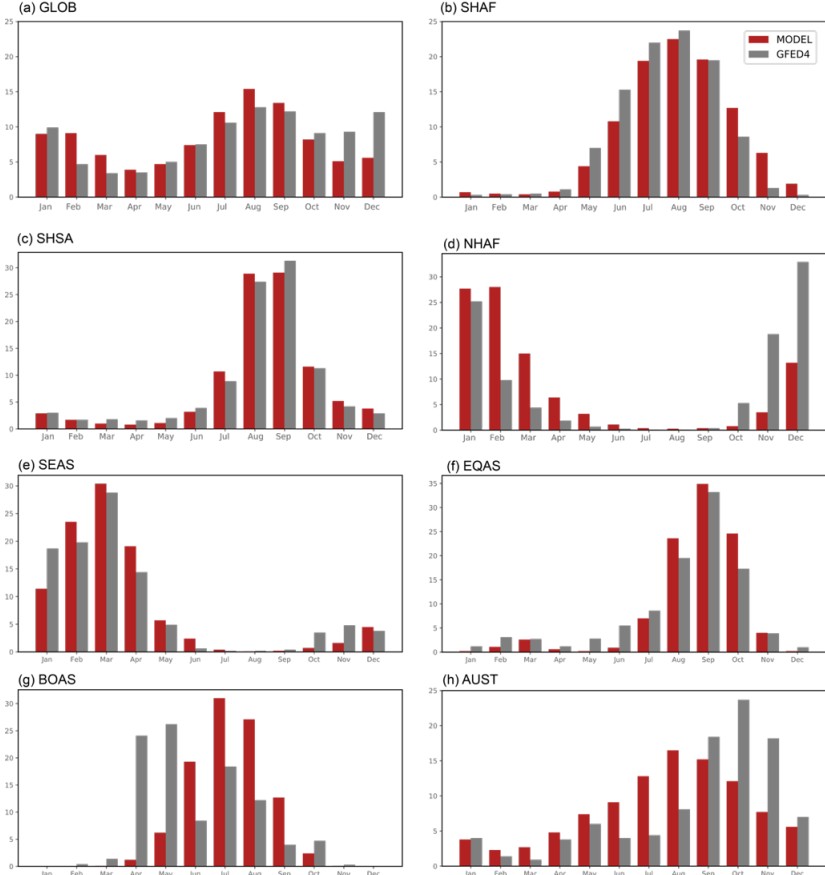

**Figure 6.** The contribution of monthly burned area to annual burned area (%) over the (a) GLOB, (b) SHAF, (c) SHSA, (d) NHSA, (e) SEAS, (f) EQAS, (g) BOAS, and (h) AUST averaged over 2000-2014 for SSiB4/TRIFFID-Fire and GFED4s





## 4.2 Fire emissions

The carbon emission in SSiB4/TRIFFID-Fire is 2.19 Pg yr$^{-1}$, higher than the estimate from GFED4s (2.07 Pg yr$^{-1}$) (Fig. 7). SSiB4/TRIFFID-Fire captures the high carbon emissions in tropical savannas, the intermediate emissions in Northern Hemisphere boreal forests, and the low emissions in humid forests and deserts with a spatial correlation of 0.72, higher than the simulation in Li et al. (2013) (0.61 compared with the GFED3). In general, the spatial distribution of carbon emissions coincides with that of the burned area: SHAF, NHAF, and SHSA are the major fire emission regions and they contribute to 65.4% of the total emission in both GFED4s and SSiB4/TRIFFID-Fire (Fig. 8a). The exception occurs in EQAS, BOAS, and BONA, where the fire emissions contribute to 11.6 % of the global emissions with only 2.5 % of the global burned area there. The regions have large areas of peatland, which contains a thick layer of soil carbon and emits several times more trace gases per unit biomass combusted than fires in savannas (van der Werf et al., 2010). As our model does not include the peat soil type, fire emissions are underestimated in these regions.

The interannual variability (IAV) of fire emissions is captured in SSiB4/TRIFFID-Fire in 7 out of 14 fire regions with a significance level $p < 0.05$ (Fig. S2). Both model and observation have shown a decrease in carbon emissions in NHAF (Fig. 8b), which contributes to the decrease in global fire emission in 2000-2014 (Fig. 7e). SSiB4/TRFFID-Fire suggests that the decline of global fire emissions starts in the 1950s, which is also found in some of the FireMIP models (Li et al., 2019). Similar to our conclusions in the IAV of burned areas, the IAV of carbon emissions in SHSA is small and is not well represented in the model (Fig. 8c).

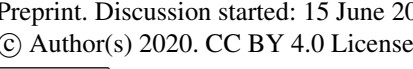

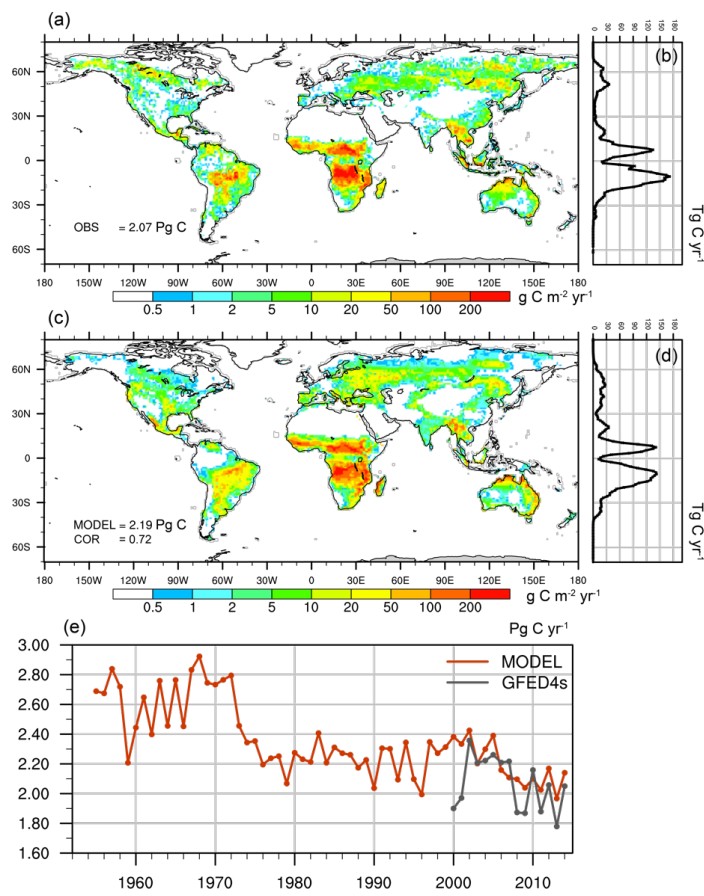

**Figure 7.** (a) Spatial distribution of annual carbon emission (g C yr$^{-1}$) averaged over 2000–2014 for GFED4s and (b) the latitudinal distribution of carbon emission (Tg C yr$^{-1}$); (c) Same as (a) but for SSiB4/TRIFFID-Fire (g C yr$^{-1}$) and (d) same as (b) but for SSiB4/TRIFFID-Fire; (e) Annual carbon emission (Pg C yr$^{-1}$) for GFED4s for 2000-2014 and for SSiB4/TRIFFID-Fire for 1948-2014




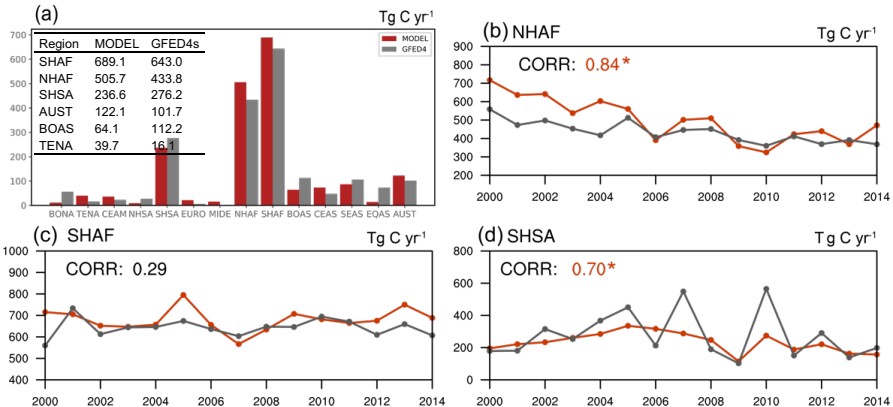

**Figure 8.** (a) Annual carbon emission averaged over 2000-2014 for GFED4s and SSiB4/TRIFFID-Fire in 14 GFED regions; (b)-(d) annual carbon emission in (b) SHAF, (c) NHAF, and (d) SHSA for 2000–2014 for GFED4s and SSiB4/TRIFFID-Fire. The "*" indicates the correlation is significant at p < 0.05.

## 4.3 GPP and PFT distribution

The simulated GPP averaged over 1982-2011 is compared to FLUXNET-MTE GPP (Jung et al., 2011) to examine the model description of fire impact on carbon and ecosystem characteristics. SSiB4/TRIFFID-Fire captures the distribution of global GPP with a spatial correlation of 0.93 (p < 0.05) (Fig. 9). The highest GPP occurs in the tropical evergreen forest and decreases with latitude in both observation and model. However, the simulated GPP has a negative bias in the Amazon tropical forest and a positive bias in tropical Africa and boreal regions. The simulated global GPP is 141 Pg C yr$^{-1}$ in SSiB4/TRIFFID-Fire, higher than the estimate, 119 Pg C yr$^{-1}$ in FLUXNET-MTE (Jung et al., 2011), but is within the range of simulated GPP in current process-based DGVMs (111 - 151 Pg C yr$^{-1}$; Piao et al., 2013). In addition, the correlation of IAV of global GPP is 0.68 (p < 0.05) between SSiB4/TRIFFID-Fire and FLUXNET-MTE. Some studies have reported that IAV of GPP is underestimated in the FLUXNET-MTE due to a smaller number of eddy covariance flux sites in tropical regions (Piao et al. 2013). Moreover, the lack of

$CO_2$ fertilization during the MTE model training may contribute to an underestimate in the

FLUXNET-MTE GPP. Improved data are needed to make a more comprehensive comparison.

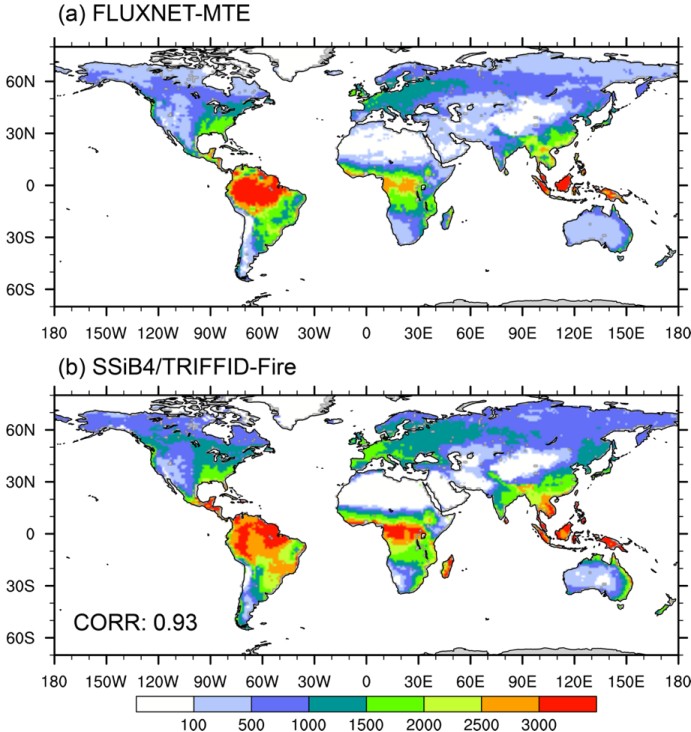

**Figure 9.** Spatial distribution of annual GPP (g C m$^{-2}$ year$^{-1}$) averaged over 1982–2011 for (a) FLUXNET-MTE and (b) SSiB4/TRIFFID-Fire

The simulation of vegetation coverage, which represents model description of biomass

allocation and influences the fuel availability and flammability in fire modeling, is evaluated

against GLC2000 (Bartholome and Belward, 2005). As the dynamic vegetation model only

includes natural PFTs, the simulated PFT fraction in one grid box is scaled using the non-

agriculture fraction from GLC2000. Overall, the vegetated areas cover 80.6 % of global land areas,

very close to the estimates from GLC2000 (80.8 %). The simulated tree cover is 34.1 %, higher

than 29.8 % in GLC2000. Most trees are located in the Amazon rainforest, tropical Africa,

Equatorial Asia, Southeast Asia, Southeast North America, and Northern Hemisphere high





latitudes (Figs. S3a-c). The simulated C3 and C4 fraction are 11.1 % and 7.5%, respectively, similar to the GLC2000 estimates (11.9 % for C3 and 7.9 % for C4). Shrubs are primarily located

in the semi-arid regions and the pan-Arctic area, and tundra is limited to the pan-Arctic area and Tibetan Plateau. SSiB4/TRIFFID-Fire is shown to capture some key processes of fire-vegetation interactions under the current climate, which is important to study fire effects on the ecosystem.

**4.4 Fire effects on ecosystem characteristics and surface properties**

In this section, we investigate long-term fire effects on ecosystem characteristics, surface

properties, and surface energy budget using the differences between FIRE-ON and FIRE-OFF (FIRE-ON minus FIRE-OFF). In SSiB4/TRIFFID-Fire, fire is found to cause a strong decrease in tree fraction on 6.14% of the global land surface, accompanied by an increase of grass cover and bare land on 3.76% and 1.25% of land surface, respectively (Figs. 10a-b). The reduction of tree cover is concentrated in Southern Africa, Northern Africa, and South America, which are

dominated by C4 savanna in FIRE-ON (Fig. S3), suggesting that fire is an important determinant of structure and functions of the savanna; otherwise it would be encroached by trees. Our results are supported by observational studies which have reported that frequent fires (especially annual and biennial) could have a marked impact on the vegetation structure and tree density, reducing trees to fire-resistant grasses with a scattering of shrubs (Higgins et al., 2007; Furley et al., 2008;

Devine et al., 2015). A long-term fire experiment in Kruger National Park, South Africa, found that fire reduced woody cover by 30-50% (Smit et al., 2010), which is generally consistent with our model results (Fig. 10a). The reduction in tree cover is associated with tree mortality and the fast re-growth of grass PFTs with space and nutrient availability after fire. Moreover, the reduction in tree cover and vegetation height is larger in wetter savanna than in drier savanna in

SSiB4/TRIFFID-Fire (Figs. 10a and 10d), which is confirmed by the Kruger National Park

experiments (Sankaran et al., 2005; Smit et al., 2010).

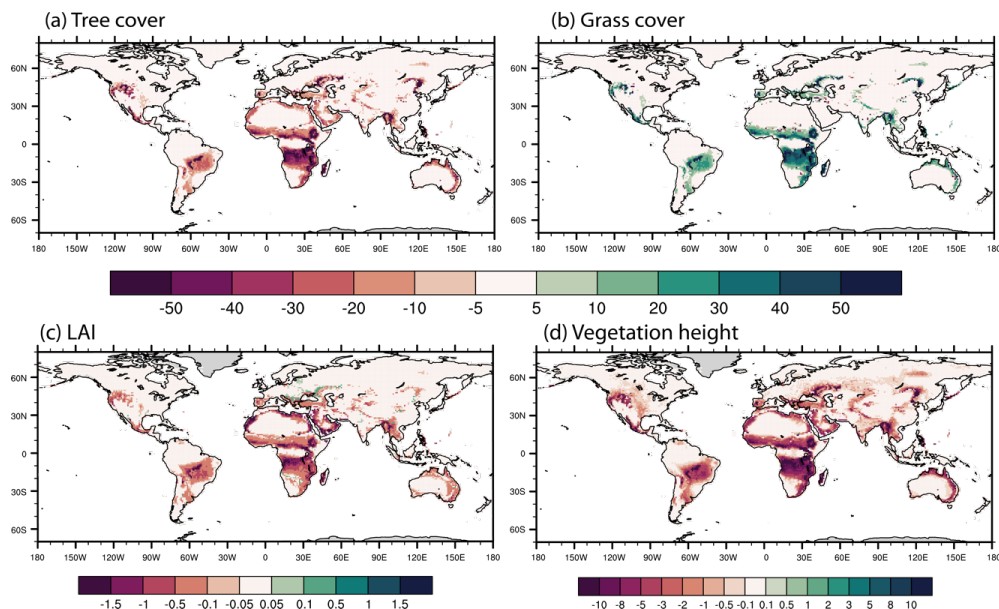

**Figure 10.** Difference in (a) tree cover (BET, NET, and BDT; %) (b) grass cover (C3 and C4; %),
(c) LAI ($m^2 m^{-2}$), and (d) Vegetation height (m) in SSiB4/TRIFFID-Fire averaged over 1948-2014
between FIRE-ON and FIRE-OFF

Based on SSiB4/TRIFFID-Fire simulations, fire causes a decrease of LAI and vegetation

height on the grid level (Figs. 10c-d), in agreement with earlier field observations across various

biomes (Shackleton and Scholes, 2000; Bond-Lamberty and Gower, 2008). The global LAI and

vegetation height are by 0.13 $m^2 m^{-2}$ and 1.27 m, respectively. The changes in ecosystem structure

modify the albedo, aerodynamic resistance, and evapotranspiration processes, which further

influences surface radiation and energy partitioning between latent heat and sensible heat fluxes.

Surface energy changes during the local fire season (DJF for NHAF, and JJA for SHAF

and SHSA) and the post-fire (rainy) season (JJA for NHAF, and DJF for SHAF and SHSA) are

shown in Fig. 11. The surface radiation is decreased in both seasons (Fig. 11 a-b) due to a





replacement of trees by grass/savanna (Figs. 10a-b), as grass PFTs normally have a higher albedo

than tree PFTs. The sensible heat flux is reduced by 4-6 W m$^{-2}$ (Figs. 11 c-d), resulting from the

decrease in surface energy and the increase in aerodynamic resistance associated with vegetation

height change (Liu et al., 2016). Our results are in agreement with observational studies on

different fire types (e.g., forest fire and savanna fire) showing that surface properties changes after

fire results in an increase in albedo (Gholz and Clark, 2002; Amiro et al., 2006b; Sun et al., 2010)

and a decrease in sensible heat (Chambers and Chapin, 2002; Liu et al., 2005; Amiro et al., 2006a;

Amiro et al., 2006b; Rogers et al., 2013).

        The change of latent heat flux (Figs. 11 e-f) varies with seasons in tropical savanna. It is

reduced in the fire season and enhanced in the rainy season in each fire regions. The decrease in

canopy evapotranspiration dominates the latent heat flux change in the fire season (DJF for NHAF,

and JJA for SHAF and SHSA); however, the reduction has been accounted for by the increase in

soil evaporation in the rainy season (JJA for NHAF, and DJF for SHAF and SHSA) (Fig. S4).

Observational studies have shown that the significant increase in soil evaporation during wet

seasons can offset the decrease in evapotranspiration and enhance latent heat after fire (Langford,

1976; Dunin, 1987; Gholz and Clark, 2002; Santos et al., 2003; Amiro et al., 2006b). The increase

is caused by the exposure of moist soil surface, the increase in surface energy that can be used for

evaporation, and the smaller surface resistance when dense plant canopy is removed by fire

(Schulze et al., 1994). Overall, an increase in temperature is found (Figs. 11g-h) due to a reduction

in surface fluxes throughout the year, consistent with the conclusions in Li et al. (2017) based on

annual results.

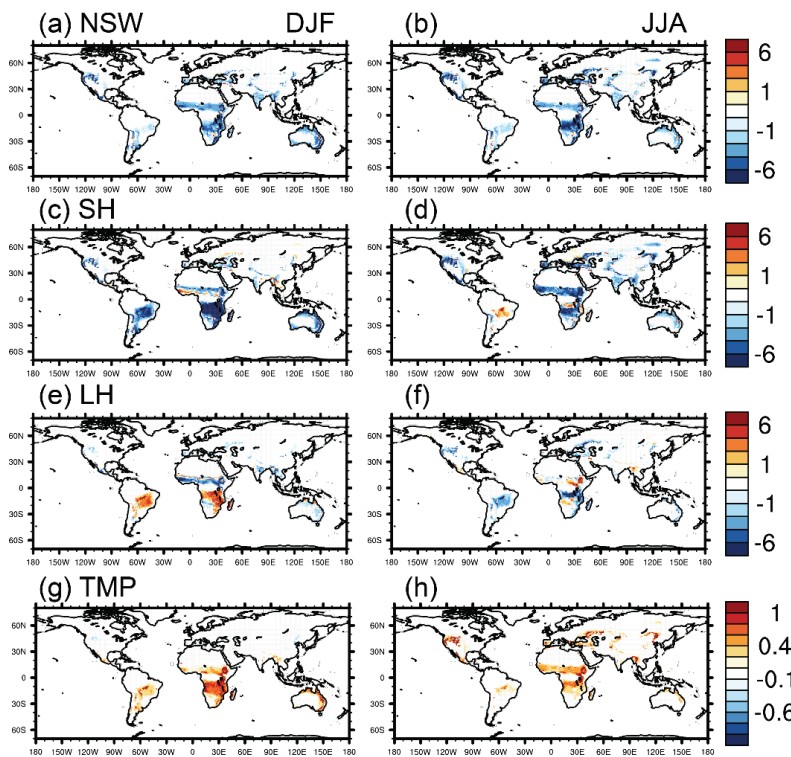

**Figure 11.** Difference in net shortwave (a, b; W m$^{-2}$), sensible heat (c, d; W m$^{-2}$), latent heat (e,f; W m$^{-2}$), and surface temperature (g, h; K) in DJF (a, c, e, and g), and JJA (b, d, f, and h) averaged over 1948-2014 between FIRE-ON and FIRE-OFF.

## 5. Conclusions

We have implemented a process-based fire model of intermediate complexity into a DGVM, SSiB4/TRIFFID, which is based on the surface water, carbon, and energy balances, as wells as the PFT competition. The high-frequency exchanges between fire model and SSiB4/TRIFFID allow vegetation dynamics and surface parameters, such as albedo and surface roughness length, to be updated every 10 days based on surface carbon balance, which are rarely applied in other fire models. Moreover, the plant production and biomass allocation are reasonably reproduced in SSiB4/TRIFFID-Fire, which have been considered to contribute to the proper burned area simulation (Forkel et al., 2019; Hantson et al., 2020). The SSiB4/TRIFFID-Fire





produces similar global burned area, major regional burned areas, and fire carbon emissions
compared to GFED4s. The model captures the decreasing trend in burned area related to human
suppression and land management and the interannual variability associated with moisture
conditions. It reasonably reproduces the global GPP and PFT distribution, which is important to
study fire effects on the ecosystem.

Fire affects the ecosystem both through the carbon combustion and vegetation mortality
during fire season and through vegetation succession during post-fire recovery which leads to
changes in ecosystem composes. Comparing the simulations with and without fire, we show that
fire has reduced global tree cover by 14.2% (6.14% of land surface), which is within the range of
tree cover decrease estimated in other DGVMs (Lasslop et al. submitted). Global LAI and
vegetation height are reduced, and surface radiation, sensible heat, and canopy evapotranspiration
are decreased as a result. There is an increase in soil evaporation. Fire has resulted in temperature
increase over most fire regions.

Our estimate of fire effects on radiation, surface fluxes, and temperature are qualitatively
consistent with Li et al. (2017) but different in the partitioning between sensible heat and latent
heat changes. The discrepancies might be attributed to the changes in vegetation distribution. As
the vegetation distribution is prescribed in Li et al. (2017), trees and grass are growing taller and
denser when the fire model is turned off. In contrast, fire has caused changes in vegetation
distribution and conversions of dominant PFTs in SSiB4/TRIFFID-Fire. Tree PFTs are spreading
in FIRE-OFF and encroaching on the tropical savanna in Southern Africa, South America, and
Northern Africa, unlike in the FIRE-ON simulation. Compared to the C4 savanna, the tree-
dominant biome has a greater contrast in vegetation height than that in LAI. Therefore, the change
in vegetation height is greater but the change in LAI is smaller in SSiB4/TRIFFID-Fire than in Li

et al. (2017), which probably causes the greater sensible heat changes in our results. Other sources of uncertainties include the differences in the partitioning between latent heat and sensible heat fluxes in land surface models, the differences in the parameterization of the evaporation processes,

and the changes due to atmospheric feedbacks, such as cloud cover and precipitation changes. As Li et al. (2017) is the only modeling study investigating the long-term fire effects on land energy budget, our simulation provides another approach that quantifies fire effects using a different land surface model with different approaches in parameterizing some land surface processes and vegetation dynamics. More studies with more land surface models/fire models/vegetation

dynamics are necessary to explore this issue further. A systematic comparison of long-term fire effects in different fire models such as the current FireMIP project (Li et al. 2019), would allow evaluation of the robustness of model simulations and identification of key uncertainties of fire impacts.

**Code availability**: The source code of fire model is archived https://zenodo.org/record/3872633#.XtXeeJ70lhE (DOI: 10.5281/zenodo.3872633)

**Author contributions:** HH, YX, and YL designed the coupling strategy between SSiB4/TRIFFID and fire model. HH conducted the simulation under the suggestions from FL and YL. HH drafted

the text and made the figures. All authors (HH, YX, FL, and YL) have contributed to the analysis methods and to the text.

**Competing interests:** The authors declare that they have no conflict of interest.



**Acknowledgments:** This work is supported by NSF Grant AGS-1419526 and AGS-1849654. The authors acknowledge Cheyenne (doi:10.5065/D6RX99HX) provided by NCAR CISL, for providing HPC resources. FL acknowledges support from the National Key R&D Program of China (2017YFA0604302 and 2017YFA0604804).

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
