# Peer review of "Modeling long-term fire impact on ecosystem characteristics and surface energy using a process-based vegetation-fire model SSiB4/TRIFFID-Fire v1.0 Huilin Huang1, Yongkang Xue1, 2, Fang Li3, and Ye Liu1"

_Geoscientific Model Development, 2020_

## Referee Comment (RC1) · Anonymous Referee #1 · 21 Jul 2020

The paper describes the adaptation of the Li et al. (2012, 2013, and 2017) fire modelling to SSiB4/TRIFFID and assessed its impact on terrestrial characteristics, carbon flux, and surface energy simulations. Although the paper itself is well written and organized, it lacks novelty in both model development and evaluation. The fire modelling is simply an adaptation of the Li et al. scheme without providing advances in model development. One may see this fire model as an inferior version of the original Li et al. fire modelling scheme implemented to CLM, because SSiB4/TRIFFID does not simulate peat soil and associated fire emissions unlike CLM. It is a pity that although the

short-temporal simulation of SSiB4/TRIFFID-Fire could have led to a better seasonal pattern of fire emissions, the paper failed to elaborate the impact. For the further development of fire modelling, the authors should ask themselves critical equations: what are the lessons learned from the FireMIP, which parts of the Li et al. scheme are still uncertain and can be improved, etc.?

The second half of the paper, the evaluation of SSiB4/TRIFFID-Fire and impact on ecosystem characteristics, is kind of repetitive of what has already been done. Li et al. and subsequent papers have already evaluated their fire modelling scheme with respect to the GFED data, and the long-term decline of the fire emissions has already been demonstrated by other studies (e.g., Arora and Menton, 2018 Nat. Commun.). From the title and Introduction, I expected that the key results would be presented in regard to the impact of fire on vegetation distribution, but the analysis was short of the level to be useful to the community. Reduction of tree cover fractions, LAI, and vegetation height, and the corresponding increase of grass cover fractions (Fig. 10) are an obvious consequence of fire occurrence, aren't they? So are the results of surface energy (Fig. 11).

I don't question authors' efforts on the SSiB4/TRIFFID-Fire development and model performance. However, to be novel among many existing studies, this paper needs to provide more rigorous analyses, such as comparison with the aerosol index from remote sensing, CO emissions from MOPIT (Yin et al., 2015 ESSD), or recent CO inversion (Zheng et al., 2019 ESSD).

In sum, this paper in current form serves very well as a SSiB4/TRIFFID model report, but not as a scientific paper. I strongly recommend to include original features either/both in model developments or/and evaluation methods.

---

## Referee Comment (RC2) · Anonymous Referee #2 · 22 Jul 2020

General Comments

This paper describes the coupling of a fire model introduced by Li et. al. (2012;2013) into a biophysical model (SSiB4) with dynamic vegetation from TRIFFID. SSiB4/TRIFFID-Fire is used to provide new analysis on changes in vegetation and surface energy budgets resulting from fire. It builds on previous work, from the Li et. al. fire model that used constant vegetation, and SSiB4/TRIFFID that used constant fire disturbance. The coupling and analysis is therefore novel, and is a useful addition to existing fire model analysis performed with other models.

[Figure]

The paper is mostly well written and the methods are clearly described. The results are generally clear and support the conclusions, with the exception of a few small areas that could benefit from further clarification as outlined below. There are some sections that could also be expanded slightly to improve the interpretation of the results. Referencing and credits is thorough and comprehensive. In general I believe the paper is well presented and would be a useful publication in GMD, once the specific comments below are addressed.

Specific Comments

Lines 85-88 – can you clarify if there are any new processes that you are looking at here, or are they the same processes as Li et al (2017) but now updated with dynamic vegetation?

Line 146 – you could cite Bistinas et al (2014) here, who looked at trends in cropland fires Bistinas, I., Harrison, S. P., Prentice, I. C. and Pereira, J. M. C.: Causal relationships versus emergent patterns in the global controls of fire frequency, Biogeosciences, 11, 5087–5101, doi:10.5194/bg-11-5087-2014, 2014.

Line 194-214 – The burning of litter is not mentioned. How is this treated in the model?

Line 269 – how long was the model spun up for? Spin-up is usually carried out over pre-industrial years to remove the effects of climate change later in the 20th Century. As the period used in this study and for the spin-up is over the mid-20th century, how is climate change mitigated in the spin-up?

Line 270 – please clarify what 'quasi-equilibrium' is and how it is measured / defined

Line 288 – please clarify if the lightning data is scaled for cloud-to-ground strikes

Line 294 – can you include more of an explanation of the treatment of agriculture in the model. The source of agricultural fraction is from GLC2000; does this represent agriculutral fraction from 2000, or earlier? It looks like the agricultural fraction is constant and does not vary over time – please state this explicitly in the description. It is

worth pointing out that the change in agricultural fraction will alter burnt area, which is not taken into account here. I partly agree that this may be implicitly included by including suppression to population density and GDP (line 371), but the spatially and temporally varying agricultural fraction will be different to population and GDP, and thus will have a different impact on the spatial pattern of burnt area.

Line 315 – It is misleading to state that there is an effect on the climate here. If I understand correctly, the climate is prescribed, but the surface temperature can vary? So you don't actually get full climate feedbacks as you would in an atmosphere or full Earth System Model?

Line 330 – Why is the magnitude of burned area underestimated in Northern Africa savannah?

Line 333 - this sentence seems contradictory. Please clarify. Do you mean that humidity counteracts high fuel, resulting in a smaller peak?

Line 410 – I can't see which part of West Africa is referred to here. Please clarify

Line 416 – It looks like this should be May instead of April. Does this change the conclusion of the sentence, which refers to the wettest month in the model?

Line 419 – we generally don't see much burning in cropland areas (see earlier reference to Bistinas et. al. 2014), so I don't see that this would be a driver of the low correlation in the two regions mentioned here.

Line 485 – nice comparison of biomass compared to GLC2000. I think it would be useful to include maps of vegetation cover without fire and from observations as well to help show the spatial pattern

Figure 4b – the axis text is very faint and difficult to read. Please change text to black instead of grey (and in all relevant figures). The caption mentions an asterisk for positive correlation, which also appears to be denoted by red text – please state this in the caption as well.

Figure 10 – what causes the changes in tree cover, LAI and vegetation height in the north Africa (Libya and Egypt, and far west across Mauritania and western Sahara)? It doesn't look like there is any burnt area simulated here.

Lines 541- 544 - I don't follow what the authors are saying here. Line 541 – what variable is being referred to here in terms of the reduction, evapotranspiration or latent heat? How is it accounted for by the increase in soil evaporation? Line 544 – this sentence is referring to the wet season, but says that latent heat increases after fire. Is there much fire in the wet season?

Technical Corrections

Line 28 – asses *the* long-term fire impact

Line 50 – *the* hydrological cycle

Line 77 – Lasslop et al submitted is now published. References can be updated throughout

Line 83 – only *the* fire model developed by Li et al

Line 87 – only annual energy *flux* changes

Line 94 – by incorporating *the* fire scheme of Li et al

Line 112 – should GPP / NPP be production / productivity rather than products?

Line 450 – model and *observations*

Line 469 – suggest reword to 'modelled impact of fire…'

Line 472 – *observations* and model

Line 524 – vegetation height area *decreased* by

Line 539 – fire *region*

---

## Author Comment (AC1) · 19 Sep 2020

The paper describes the adaptation of the Li et al. (2012, 2013, and 2017) fire modelling to SSiB4/TRIFFID and assessed its impact on terrestrial characteristics, carbon flux, and surface energy simulations. Although the paper itself is well written and organized, it lacks novelty in both model development and evaluation. The fire modelling is simply an adaptation of the Li et al. scheme without providing advances in model development. One may see this fire model as an inferior version of the original Li et al. fire modelling scheme implemented to CLM, because SSiB4/TRIFFID does not simulate peat soil and associated fire emissions unlike CLM. It is a pity that although the short-temporal simulation of SSiB4/TRIFFID-Fire could have led to a better seasonal pattern of fire emissions, the paper failed to elaborate the impact. For the further development of fire modelling, the authors should ask themselves critical equations: what are the lessons learned from the FireMIP, which parts of the Li et al. scheme are still uncertain and can be improved, etc.?

We thank the reviewer's appreciation on the paper writing and structure yet we respectively disagree with the reviewer's opinions that our model lacks novelty in model development and evaluation and use the model development as a criterion to assess the paper. In the following paragraphs, the reviewers' comments are presented in black font and our point-by-point responses are in blue.

The paper's title is "Modeling long-term fire impact on ecosystem characteristics and surface energy using a process-based vegetation-fire model", which focuses on the long-term fire impact. How can use "lacking novelty in model development and validation" as an issue to judge this paper? For instance, most land models, including CLM, use Farquhar-Berry-Sellers-Collatz module for their photosynthesis process, which was originally developed in the SIB. Can we use "lacking novelty in model development and validation" as a justification to exclude CLM from investigating any terrestrial carbon budget and impacts?

As a matter of fact, in this work, after we incorporated Li fire scheme in the SSiB4/TRIFFID, we have worked with the original model developer and co-author of this paper, Dr. Fang Li, to improve the fuel combustibility parameterization, and to develop a comprehensive presentation of fire effects on vegetation distribution. After these developments, the newly-developed SSiB4/TRIFFID-Fire model has been applied to provide the first estimates of fire effect on surface energy on the seasonal scale, which is the focus of this paper. We did not emphasize this aspect is because we want to focus on the "impact" as shown in the paper title. Based on the reviewer's comments, we have clarified the novelty and significance of our work in the revised version, including:

(1) Model development: Before our work, SSiB4/TRIFFID assumed carbon disturbance caused by fire to be PFT-dependent constants without spatial and temporal changes (Cox, 2001; Liu et al., 2019). In reality, fire has high temporal and spatial variability. When we

introduced a fire scheme in, the SSiB4/TRIFFID-Fire can reproduce observed fire well. Meanwhile, the adaptation of Li fire scheme produces a better simulation in biophysical variables, such as LAI, GPP, and fire-induced vegetation disturbance compared to the original SSiB4/TRIFFID.

About model modification, we re-parameterized the fuel combustibility that affected fire occurrence and spread. In Li fire scheme, this factor is parameterized as a function of root-zone soil moisture potential factor $\beta$ (a model-dependent variable used to calculate transpiration in CLM; (Li and Lawrence, 2017)). $\beta$ was changed much from CLM4 to CLM5. Its narrow range in CLM5 also led to fire model too sensitive to drought. In SSiB4/TRIFFID-Fire, we identified the root-zone soil moisture $\theta$ was the best variable to describe fuel combustibility and set upper ($\theta_{up}$) and lower thresholds ($\theta_{low}$) as PFT-dependent. As soil moisture $\theta$ is a commonly-used variable in land surface models, our modification could provide an alteration and important reference when they are coupled with Li fire model. Besides, we adjusted parameters in fire spread rate, fuel availability, fire combustion completeness, and fire-induced mortality.

About coupling strategy, when fire model is used in CLM, the dynamic vegetation component is **inactive** and the vegetation fractional coverage is prescribed using satellite-based land cover. Therefore, Li et al. (2012; 2013; 2017) cannot describe fire impact on vegetation cover, which has been observed in many site-level studies (Moreira, 2000; Furley et al., 2008; Smit et al., 2010). In SSiB4/TRIFFID-Fire, we set the fire-related carbon loss due to combustion and post-fire mortality transferred to changes of PFT fraction based on carbon balance. In this way, SSiB4/TRIFFID-Fire is able to describe vegetation removal and bare soil exposure after fire disturbance. The model also considers the different recovery of grass/tree during post-fire seasons based on their accumulated NPP and inter-PFT competition. As a fire modeler, this type of works is not trivial but significant in the model development. The land model has evolved for forty years. Its development trajectory shows this is what our approach is. Every model has its strong and weak points. If everyone uses their own strong point to deny other models' work, then there will be no any development in land model.

(2) Model evaluation: After comprehensive evaluation of fire simulations in SSiB4/TRIFFID-Fire, we found SSiB4/TRIFFID-Fire could reasonably reproduce the total burned area as well as the temporal and spatial variability. Especially, the seasonality and interannual variabilities are reproduced well in SSiB4/TRIFFID-Fire, which is a great challenge in current FireMIP models (Li et al., 2019; Hantson et al., 2020). Per the reviewer's comment, we have added a new figure (Fig. 9) to show the comparison between observed and simulated monthly CO in 2000-2014 over Africa. Previously fire modeling studies have never evaluated the spatial distribution of fire regimes at monthly scale before.

(3) Model application: The focus of this paper is on the long term impact. Earlier studies only quantified the annual impacts of fire (mostly focused on the fire effects on carbon) or did not consider the role of fire on vegetation distribution. Using SSiB4/TRIFFID-Fire, our work, for the first time, quantified the fire impact on seasonal energy budgets with the

consideration of fire impact on vegetation coverage. We found the conversions of dominant PFTs in savannah result in a larger reduction in vegetation height (49%) than in LAI (12.5%). The resultant change in aerodynamic resistance could significantly reduce the sensible heat (the magnitude of change can be comparable to latent heat change), which further advances Li et al. (2017)'s conclusion. Moreover, we found the vegetation cover loss can cause an increase in soil evaporation in the post-fire rainy season, which could outcompete the reduction in canopy transpiration and thus results in an increase of latent heat on seasonal scale.

We believe these achievements should be considered as an essential development in simulation of both fire behavior and vegetation dynamics, which is in line with what is expected for the scope of Geoscientific Model Development.

The second half of the paper, the evaluation of SSiB4/TRIFFID-Fire and impact on ecosystem characteristics, is kind of repetitive of what has already been done. Li et al. and subsequent papers have already evaluated their fire modelling scheme with respect to the GFED data, and the long-term decline of the fire emissions has already been demonstrated by other studies (e.g., Arora and Menton, 2018 Nat. Commun.).

The most recent FireMIP ensemble analyses have shown that: there are many aspects in the fire modeling need to be improved. For instance, Hantson et al. (2020) point out that "The FireMIP models generally do not simulate the timing of peak fire occurrence accurately and tend to simulate a fire season longer than observed" and "The models are unable to represent interannual variations in burnt area". To improve the seasonal and interannual variations in fire simulation will be the key goals for future model development. In this study, simulation of fire seasonality is an important focus. Our results show that SSiB4/TRIFFID-Fire captures the fire seasonality over the globe and in a number of large fire regions, such as, Southern Hemisphere South Africa, Southern Hemisphere South America, Southeast Asia, and Equatorial Asia. Based on Dr. Li's observation, our model produces the most realistic fire seasonal simulation among all fire models in FireMIP. The representation of fire-vegetation feedback, which influences the fuel accumulation and removal should have a positive influence on this improvement. Our approach could provide a useful approach for other FireMIP models to improve their ability in modeling the fire seasonal and interannual variability.

The assessment of model uncertainty is a very important aspect of model development. The land surface model and its embedded dynamic vegetation model simulates the meteorological and fuel conditions to feed the fire scheme which presents the fire behavior. It is expected that a fire scheme would not generate the same results when it is coupled into different land surface schemes. When the SPITFIRE fire model is firstly developed in LPJ-GUESS (Thonicke et al., 2010) and then coupled to LPJ (LPJ-LMfire; Pfeiffer et al., 2013; GMD), JSBACH (JSBACH-SPITFIRE; Lasslop et al., 2014; JAMES), and ORCHIDEE (ORCHIDEE-SPITFIRE; Yue et al., 2014; GMD), the model validation and evaluation were conducted for all basic variables such as burned area and carbon emissions. As we made modifications in the fire model and we have a very different

coupling strategy between fire and SSiB4/TRIFFID, we just follow a normal community approach to conduct a similar validation and evaluation.

From the title and Introduction, I expected that the key results would be presented in regard to the impact of fire on vegetation distribution, but the analysis was short of the level to be useful to the community. Reduction of tree cover fractions, LAI, and vegetation height, and the corresponding increase of grass cover fractions (Fig. 10) are an obvious consequence of fire occurrence, aren't they? So are the results of surface energy (Fig. 11).

The reviewer has a very strange logic here. He/she claims the fire would reduce vegetation is an obvious thing, so the fire impact on vegetation has no research value. Following this logic, fire impact on emission is also an obvious thing. Based on this reviewer, why do we need to do any fire research? Currently, there is historical destruction due to wildfire in western U.S. Proper assessment of the fire damage on the ecosystem is imperative and has significant societal, economic, and ecological consequences. The reviewer claims it is not useful for "community". Which community the reviewer refers to? We are disappointed the reviewer uses unprofessional language and shows no respect to other's work.

It should be pointed out that fire impact on the ecosystem and surface energy (biophysical fire effects) is far from well understood although the fire effects have been studied since the 1990s. The long-term fire observations were conducted to understand the fire effect in grass-tree interactions, yet the results are reported on site-levels and the experiments are conducted in very limited regions (Furley et al., 2008). As the fire impact varies considerably with fire intensity and seasonality, climate, soil nutrients, and herbivory (Scogings and Sankaran, 2020), these results are only representative on the local scale.

Therefore, the fire-coupled DGVMs are developed to describe the climate potential of the global ecosystem without fire disturbance and to quantify fire impact at continental and even global scale. Bond et al. (2005) is the first modeling study to investigate the long-term fire impact on global vegetation. The author concluded that the forest cover would double in a world without fire. In the most updated fire-coupled DGVMs, the long-term fire impact on tree area is much lower (35% reduction in forest cover and 10% in tree cover averaged over 4 models; Lasslop et al. 2020), indicating there are large discrepancies in our understanding of how large the impact fire can exert on the ecosystem.

In this paper, we provide an independent estimate of biophysical fire effects using SSiB4/TRIFFID-Fire, which has a different coupling strategy, fire-vegetation feedback, and land surface model compared to any model used in Lasslop et al. (2020). Our results show 12.6% reduction in tree cover. Moreover, our work provides the first quantification in the fire impact on energy budgets on seasonal scale with the consideration of fire impacts on vegetation distribution. We found the conversions of dominant PFTs over the South American and African savannah have caused a reduction of areal-averaged LAI and vegetation height by 0.52 $m^2$ $m^{-2}$ (12.5%) and 5.76 m (49.1%), respectively. The

change in vegetation height significantly decreases the roughness length and increases aerodynamic resistance, which probably results in the greater change in sensible heat in our model as compared to Li et al. (2017). We believe our results could provide important insights in the fire effects studies for future fire model applications.

I don't question authors' efforts on the SSiB4/TRIFFID-Fire development and model performance. However, to be novel among many existing studies, this paper needs to provide more rigorous analyses, such as comparison with the aerosol index from remote sensing, CO emissions from MOPIT (Yin et al., 2015 ESSD), or recent CO inversion (Zheng et al., 2019 ESSD).

We agree with the reviewer that model validation with the most updated datasets is necessary. Per reviewer's comments, we have compared the simulated CO emission with observations from Zheng et al. (2019). The simulated global CO emission is 433.7 Tg year$^{-1}$ in 2000-2014, very close to the observations (434.0 Tg year$^{-1}$) with a spatial correlation of 0.74. The comparison between observed and simulated monthly CO in 2000-2014 over African Continent is added in the new Fig. 9, which is the first evaluation of the spatial distribution of monthly fire regimes in fire modeling studies. We conclude that SSiB4/TRIFFID-Fire provides very good estimate of peak CO emissions during fire season in JJAS in South Africa. It also captures the peak CO emission in DJF in Northern Hemisphere Africa, with slightly overestimate in January and February. The seasonality of CO emission broadly follows that of burned area, which further demonstrates that our model has shown promising results in seasonal fire simulations. The relevant analyses are added in Line 491-Line 502

[Figure]

Monthly CO emission in 2000-2014 in Africa from (a) Zheng et al. (2019) and (b) SSiB4/TRIFFID-Fire

In sum, this paper in current form serves very well as a SSiB4/TRIFFID model report, but not as a scientific paper. I strongly recommend to include original features either/both in model developments or/and evaluation methods.

Although we could not agree with the reviewer that our paper lacks novelty in model development and evaluation, we agree that the paper can be restructured to highlight our novel features. We have made the following major changes in response to the reviewer's comments. We hope the revised manuscript can better present our model development to readers from a broader background:

1. The model development in fuel combustibility parameterization and coupling strategy to include fire effects on vegetation cover has been clearly pointed out in Lines 179-184 and Lines 231-234. Section 2.3.2 (Including the fire effect on the carbon pool) in the previous draft has been moved to section 2.2.4 as a part of fire model and modifications.

2. The comparison between observed and simulated monthly CO in 2000-2014 is added in Fig. 9, which is the first evaluation of the spatial distribution of monthly fire regimes in fire modeling studies. The relevant discussion is added in Lines 491-502.

**Reference:**

[revised manuscript text omitted]

---

## Author Comment (AC2) · 19 Sep 2020

This paper describes the coupling of a fire model introduced by Li et. al. (2012;2013) into a biophysical model (SSiB4) with dynamic vegetation from TRIFFID. SSiB4/TRIFFID-Fire is used to provide new analysis on changes in vegetation and surface energy budgets resulting from fire. It builds on previous work, from the Li et. al. fire model that used constant vegetation, and SSiB4/TRIFFID that used constant fire disturbance. The coupling and analysis is therefore novel, and is a useful addition to existing fire model analysis performed with other models.

The paper is mostly well written and the methods are clearly described. The results are generally clear and support the conclusions, with the exception of a few small areas that could benefit from further clarification as outlined below. There are some sections that could also be expanded slightly to improve the interpretation of the results. Referencing and credits is thorough and comprehensive. In general I believe the paper is well presented and would be a useful publication in GMD, once the specific comments below are addressed.

We thank the reviewer for pointing out the importance of our work and providing constructive comments to improve our analyses. We have a better understanding of the model evaluation and application after exchanges with the reviewer. Below we provide a point-by-point response to the reviewer's comments. In the following paragraphs, the reviewer's comments are in black font and our responses are in blue.

Specific Comments

Lines 85-88 – can you clarify if there are any new processes that you are looking at here, or are they the same processes as Li et al (2017) but now updated with dynamic vegetation?

Our work has 2 major differences compared to Li et al. (2017) although we both look at fire impact on the energy budget. First, with the dynamic vegetation included, we describe the vegetation loss during fire season and the recovery of different PFTs decided by the competition strategy during the post-fire rainy season. This fire effect is reflected on the seasonal scale. Moreover, the long-term fire impact can influence grass-tree interactions and change vegetation composition in SSiB4/TRIFFID-Fire. These processes are not described in Li et al. (2017) as it applies prescribed land cover. The original introduction did not clearly point out what we are looking for in addition to Li et al. (2017). It has been added in Lines 86-93

Line 146 – you could cite Bistinas et al (2014) here, who looked at trends in cropland fires Bistinas, I., Harrison, S. P., Prentice, I. C. and Pereira, J. M. C.: Causal relationships versus emergent patterns in the global controls of fire frequency, Biogeosciences, 11, 5087–5101, doi:10.5194/bg-11-5087-2014, 2014.

Cited. Thank you.

Line 194-214 – The burning of litter is not mentioned. How is this treated in the model?

The carbon cycle in the current SSiB4/TRIFFID version does not explicitly represent the litter carbon storage and decomposition. The litter fall due to turnover of leaf, wood, and root, and fire disturbance is directly turned into soil carbon. Previous studies have shown that litter carbon and woody debris account for about 25% of aboveground biomass for global forest (Pan et al., 2011) and accounts for about 30% for savanna (de Oliveira et al., 2019). Based on these studies, we provide a rough estimate of aboveground litter proportional to the aboveground biomass. The fire emission from litter is calculated in the similar way as other aboveground carbon, i.e., a PFT-dependent combustion completeness factor is used to calculate the burning of litter produced by different PFTs:

Table PFT-dependent combustion completeness factors for litter ($CC_{litter}$)

| PFT | $CC_{litter}$ |
| --- | --- |
| BET | 0.30 |
| NET | 0.55 |
| BDT | 0.30 |
| C3 grasses | 0.80 |
| C4 plants | 0.80 |
| Shrubs | 0.80 |
| Tundra | 0.80 |

The description of the burning of litter is added in Lines 212-216. We acknowledge that the parameterization of the burning of litter isn't perfect and is mainly used to match the observed carbon emission. The next generation of SSiB (SSiB5/TRIFFID-DayCent-SOM) couples SSiB/TRIFFID with DayCent-SOM, which describes the full processes of litter accumulation and decomposition, constrained by nitrogen availability (Parton et al., 1994). Our next goal is to include fire model in the SSiB5/TRIFFID-DayCent-SOM, and an explicit parameterization of litter combustion will be updated in the new version. We add this discussion to the revised manuscript in the future model development (Lines 654-660).

Line 269 – how long was the model spun up for? Spin-up is usually carried out over pre-industrial years to remove the effects of climate change later in the 20th Century. As the period used in this study and for the spin-up is over the mid-20th century, how is climate change mitigated in the spin-up?

The spin-up run for fire simulations are conducted for 100 years using SSiB4/TRIFFID-Fire. It is added in Line 278

The spin-up runs are widely used in climate models to remove the effects of climate change. In this study, however, the goal of our spin-up simulations is to generate a quasi-equilibrium vegetation distribution as initial conditions for our model transient simulation from 1948, with the forcing data from Sheffield et al. (2006). Otherwise, the vegetation will have a very strong spin up in the first several decades' integration.

Line 270 – please clarify what 'quasi-equilibrium' is and how it is measured / defined

To clarify this, we have added "Following Liu et al. (2019), the quasi-equilibrium status defined as the rate of relative change in fractional coverage of all PFTs is less than 2 % over the last 10 years of simulation" in Lines 282-284. In this experiment, we found 100-year spin-up time is sufficient for our model to reach quasi-equilibrium.

Line 288 – please clarify if the lightning data is scaled for cloud-to-ground strikes

The lightning data we use from NASA LIS/OTD v2.2 is the total lightning flashes and we apply the scale factor ($\frac{1}{5.16+2.16 \cos [3min(60,\lambda)]}$; $\lambda$: latitude) to derive the cloud-to-ground lightning following Prentice and Mackerras (1977). We add the clarification in the model description in Line 154-160

Line 294 – can you include more of an explanation of the treatment of agriculture in the model. The source of agricultural fraction is from GLC2000; does this represent agricultural fraction from 2000, or earlier? It looks like the agricultural fraction is constant and does not vary over time – please state this explicitly in the description. It is worth pointing out that the change in agricultural fraction will alter burnt area, which is not taken into account here. I partly agree that this may be implicitly included by including suppression to population density and GDP (line 371), but the spatially and temporally varying agricultural fraction will be different to population and GDP, and thus will have a different impact on the spatial pattern of burnt area.

The GLC2000 agriculture fraction applied in our model represents the agricultural fraction for the year 2000 (Bartholome and Belward, 2005). It has been explicitly pointed out in Line 309. We agree with the reviewer that the interannual variation of cropland distribution influences the spatial and temporal variations of burned area beyond GDP and population effects. It should be considered in future model development. Following the reviewer's suggestion, we have discussed the potential biases caused by the constant agricultural fraction in the conclusions and discussion (Lines 650-654)

Line 315 – It is misleading to state that there is an effect on the climate here. If I understand correctly, the climate is prescribed, but the surface temperature can vary? So you don't actually get full climate feedbacks as you would in an atmosphere or full Earth System Model?

Thanks for point this out. The SSiB4/TRIFFID-Fire is run in offline mode where Sheffield et al. (2006) provide the lowest level atmospheric forcing (atmosphere lowest-level temperature, specific humidity, surface pressure …etc.) to the land model. Therefore, it is inappropriate to say that "SSiB4/TRIFFID-Fire is applied to assess the long-term fire effect on … climate". We have corrected it as "After model validation, SSiB4/TRIFFID-Fire is applied to assess the long-term fire effect on the ecosystem and surface energy budget using the differences between the FIRE-ON and FIRE-OFF."

Line 330 – Why is the magnitude of burned area underestimated in Northern Africa savannah?

The fire parameters in the model are set to generally reproduced the global-average burned area and spatial distribution. With this global uniformed parameters, SSiB4/TRIFFID-Fire simulate a global burned area very close to GFED4s estimates with a correlation of 0.8. However, it underestimates burned area in many regions (e.g., Northern Africa savannah and Australia), and overestimates in others (e.g., South America and temperature North America).

Regarding the underestimate of burned area in North Africa savannah, it may due to the difference in vegetation species with other parts of the world (although they are categorized in the same PFTs), which influences the description of fuel availability and combustibility. Besides, the sparse meteorological stations in Northern Africa savannah may influence the accuracy in climate forcings. For a better simulation of fires in Northern Africa savannah, we need to have more information on fuel conditions, how climate controls fire, and how human activities influence fire suppression there.

Line 333 - this sentence seems contradictory. Please clarify. Do you mean that humidity counteracts high fuel, resulting in a smaller peak?

The humidity acts as a suppression factor for fire ignitions in Boreal Asia and Boreal North America but the high level of aboveground fuel facilitates fire occurrence and spread. Therefore, the Boreal region has an intermediate burned area which is lower than savanna but higher than the desert. This sentence has been revised in Line 352-355 in the latest manuscript.

"Another burned area peak occurs around 50º N in Boreal Asia (BOAS) and Boreal North America (BONA). The humid climate there suppresses fire ignition yet the high level of aboveground biomass and a lack of human suppression facilitate fire occurrence and spread, which results in an intermediate burned area in the boreal regions."

Line 410 – I can't see which part of West Africa is referred to here. Please clarify

The word of "West Africa" may be confusing — we meant to refer to the whole Northern Hemisphere Africa shown in the orange box below. The correlation of monthly burned area between the observed and simulated fire in Northern Hemisphere Africa is lower

than 0.5 and is not as good as that in the Southern Hemisphere Africa. We have changed the word to "Northern Hemisphere Africa" in the revisions.

[Figure]

Temporal correlation of monthly burned area in West Africa

Line 416 – It looks like this should be May instead of April. Does this change the conclusion of the sentence, which refers to the wettest month in the model?

Sorry for the confusion. The largest burned area is observed in May for the whole Boreal Asia region, yet the West Siberia has the largest fire in April, which is not captured in SSiB4/TRIFFID-Fire and thus causes the apparent underestimate in April (figure below). A similar scenario is also found in May, with slightly more burned area simulated in West Siberia. Both underestimates in April and May are caused by the wet bias from the forcing. To avoid confusion, we have modified the sentence as follow (Lines 435-438):

"In BOAS, the first fire season occurs in April and May. The observed fire season is not captured in SSiB4/TRIFFID-Fire as the model underestimates the burned area in western Siberia due to too wet moisture conditions that come from the high precipitation and specific humidity in the forcing data"

[Figure]

April and May burned fraction (%) in BOAS in GFED4

[Figure]

April and May burned fraction (%) in BOAS in SSiB4/TRIFFID-Fire

Line 419 – we generally don't see much burning in cropland areas (see earlier reference to Bistinas et al. 2014), so I don't see that this would be a driver of the low correlation in the two regions mentioned here.

Bistinas et al. (2014) has shown that the croplands normally limit burned area due to landscape fragmentation. On the other hand, fires can be ignited by humans on cropland to clear the crop residues in many regions. These fires are generally small in size and below the detection limit of MODIS. Therefore, they are not included in previous GFED data (e.g. GFED3.1 used in Bistinas et al., (2014)). By applying the most updated GFED data which explicitly considers small fires, recent studies have shown that the small fires in eastern China and the middle east U.S. are related with agricultural fires (Randerson et al., 2012; van der Werf et al., 2017).

Take China for example, Xia et al. (2013) show that the fire peak in North China Plain and Mid-eastern China in May and June are likely to relate with the harvest of winter wheat, in which the fires are used to clear the crop residues. In areas south of 30°N, fire peak occurs in winter, mainly associated with the sugar-cane harvest in November–April. However, current SSiB4/TRIFFID-Fire version does not include the agriculture fire, which we think may contribute to the missing fire peak in eastern China. The references on the seasonality of agricultural fire are added in Line 442.

[Figure]

Monthly burned fraction (%) in eastern China in GFED4s

Line 485 – nice comparison of biomass compared to GLC2000. I think it would be useful to include maps of vegetation cover without fire and from observations as well to help show the spatial pattern

Very good suggestion. We agree that it would be useful to compare the vegetation cover in FIRE-ON with the vegetation cover in FIRE-OFF and validate it with observations. We have added a new Fig. 10 to compare the tree and grass fractional coverage between observations, FIRE-ON, and FIRE-OFF. We find that the FIRE-ON simulation has generally reproduced the observed tree cover and grass cover while the FIRE-OFF overestimated tree covers in Northern African, Southern African, and South American savanna, indicating that fire is an important factor for the existence of savanna biome. The evaluation of simulated vegetation cover in FIRE-ON using observations is discussed Lines 515-519. The original validation of GPP (Figure 9 in the previous manuscript) is moved to Fig. S4 in the supplementary material.

When fire is turned off (FIRE-OFF), the tree cover expands in the tropical savanna where the climate allows trees to approach canopy closure such that grasses may be effectively excluded (Fig. 10e). Meanwhile, the grass cover shrinks to the southern part of Southern Africa savanna and the northern part of Northern Africa savanna (Fig. 10f) where tree growth may be constrained by environmental conditions.

Our results are consistent with the long-term fire experiments which reported that fire strongly affected vegetation structure, lowering the proportions of trees to fire-resistant grasses and shrubs (Higgins et al., 2007; Furley et al., 2008; Devine et al., 2015), and

that fire impact is more remarked in wetter savanna than in drier savanna (Moreira, 2000; Sankaran et al., 2005). However, the long-term fire experiments were conducted in very limited regions and mostly focused on site-level fire impact (Furley et al., 2008). The assessment of continental and global fire impact on vegetation and carbon can only be achieved by fire-coupled DGVMs. After appropriate validation of fire effects on the local scale, SSiB4/TRIFFID-Fire can be used as an effective tool to describe the climate potential of the ecosystem without fire disturbance and to quantify fire impact on the global scale. The above discussion is added in Lines 555-574 to expand our interpretation of results.

[Figure]

Figure. 10 The spatial distribution of trees in (a) GLC2000 (c) FIRE-ON, and (e) FIRE-OFF and the spatial distribution of grasses in (b) GLC2000 (d) FIRE-ON, and (f) FIRE-OFF in 2000

Figure 4b – the axis text is very faint and difficult to read. Please change text to black instead of grey (and in all relevant figures). The caption mentions an asterisk for positive correlation, which also appears to be denoted by red text – please state this in the caption as well.

The labels in Fig. 4b, Fig. 6, and Fig. 8a have been changed to black color. Both red color and "*" denote the correlation is significant. The statement is added in Fig.4 and Fig. 8.

Figure 10 – what causes the changes in tree cover, LAI and vegetation height in the north Africa (Libya and Egypt, and far west across Mauritania and western Sahara)? It doesn't look like there is any burnt area simulated here.

Sorry, we just found that we inadvertently linked to a wrong data set when we draw this figure. We are terribly sorry for this mistake and the confusion that caused. Thanks for pointing out this. This figure has been corrected using the correct data set (and the relevant number in the manuscript is also changed) and it turns out that there are not many changes in the north Africa (Libya and Egypt, and far west across Mauritania and western Sahara). We also checked other figures and they do not have this error.

Besides, we have changed the study period to 2000–2014 in Fig. 11-12 to be consistent with the period used in burned area validation.

[Figure]

Difference in (a) tree cover (BET, NET, and BDT; %) (b) grass cover (C3 and C4; %), (c) LAI ($m^2$ $m^{-2}$), and (d) Vegetation height (m) in SSiB4/TRIFFID-Fire averaged over 2000-2014 between FIRE-ON and FIRE-OFF

Lines 541- 544 - I don't follow what the authors are saying here. Line 541 – what variable is being referred to here in terms of the reduction, evapotranspiration or latent heat? How is it accounted for by the increase in soil evaporation? Line 544 – this sentence is referring to the wet season, but says that latent heat increases after fire. Is there much fire in the wet season?

Sorry for the confusion. The "reduction" referred to the decrease in canopy evapotranspiration that was mentioned in the last sentence. There was a typo in Line 541, "accounted for" should be "offset by".

In SSiB4, the grid average latent heat consists of canopy evapotranspiration, canopy interception and soil evaporation. The canopy interception normally plays a minor role.

During fire season, the canopy transpiration is decreased as fire-induced vegetation canopy loss. The soil evaporation does not change too much as there is not much surface water to evaporate. Therefore, the grid average latent heat is decreased. During wet season (after the fire season), the canopy transpiration is still decreased as the vegetation has not recovered from the fire. However, the exposure of bare soil has produced more soil evaporation. Therefore, the latent heat can be increased due to enhanced soil evaporation. The sentences have been reworded in Line 596 to Line 604 in the revised draft.

Technical Corrections
Line 28 – asses *the* long-term fire impact
Done

Line 50 – *the* hydrological cycle
Done

Line 77 – Lasslop et al submitted is now published. References can be updated throughout
Thank you. Lasslop et al. (2020) has been added and updated throughout.

Line 83 – only *the* fire model developed by Li et al
Done

Line 87 – only annual energy *flux* changes Line 94 – by incorporating *the* fire scheme of Li et al
Done

Line 112 – should GPP / NPP be production / productivity rather than products?
Yes. It has been corrected.

Line 450 – model and *observations* Line 469 – suggest reword to 'modelled impact of fire. . .'
Done

Line 472 – *observations* and model Line 524 – vegetation height area *decreased* by Line 539 – fire *region*
The words have been corrected. Thank you.

Reference:

[revised manuscript text omitted]

---

## Author Response (AR2)

In my view the authors have addressed all of the review comments thoroughly, and the paper is improved as a result of the additional analysis and figures.

We thank the reviewers' constructive comments in the interactive discussion that help us improve the paper structure and analyses. We have addressed the technical corrections listed below:

I spotted two minor typos in the new text which could be addressed prior to publication:

Line 231: Change 'When Li et al fire model is coupled' to 'When the Li et al. fire model is coupled'
Done.

Line 624: 'areal-averaged LAI' should be 'area-averaged'?
Corrected. Thank you.